# Biosynthesis of strychnine

Benke Hong[1], Dagny Grzech[1], Lorenzo Caputi[1], Prashant Sonawane[1], Carlos E. Rodríguez López[1], Mohamed Omar Kamileen[1], Néstor J. Hernández Lozada[1], Veit Grabe[2] & Sarah E. O'Connor[1✉]

Strychnine is a natural product that, through isolation, structural elucidation and synthetic efforts, shaped the field of organic chemistry. Currently, strychnine is used as a pesticide to control rodents[1] because of its potent neurotoxicity[2,3]. The polycyclic architecture of strychnine has inspired chemists to develop new synthetic transformations and strategies to access this molecular scaffold[4], yet it is still unknown how plants create this complex structure. Here we report the biosynthetic pathway of strychnine, along with the related molecules brucine and diaboline. Moreover, we successfully recapitulate strychnine, brucine and diaboline biosynthesis in *Nicotiana benthamiana* from an upstream intermediate, thus demonstrating that this complex, pharmacologically active class of compounds can now be harnessed through metabolic engineering approaches.

Strychnine—a complex monoterpene indole alkaloid—was isolated in 1818 from the seeds of *Strychnos nux-vomica* (poison nuts)[5], which were used in traditional medicine in China and South Asia. Currently, strychnine is used as a pesticide[1] because of its neurotoxicity, which is mediated by high-affinity binding to the glycine receptor[2,3]. Approximately 130 years after its isolation, the structure of strychnine was independently elucidated by Robinson in 1946 (refs. [6,7]) and Woodward in 1947 (ref. [8]). Robinson noted that 'for its molecular size, it is the most complex substance known'[9]. For centuries, strychnine had a large role in the field of chemistry through its isolation, structural elucidation and synthesis (Supplementary Fig. 1). Its polycyclic architecture inspired chemists to develop new synthetic transformations and strategies, and ultimately led to a number of total syntheses[4] since the first seminal total synthesis in 1954 (ref. [10]). Surprisingly, it is still unknown how plants create this complex structure. Here we report the biosynthetic pathways of strychnine, brucine and diaboline.

A partial biosynthetic hypothesis of strychnine was proposed in 1948 (ref. [11]), which was substantiated by feeding studies of radioisotope-labelled substrates in *S. nux-vomica*[12–15]. These labelling studies demonstrated that, like all monoterpene indole alkaloids, strychnine **10** originates from tryptophan and geranyl pyrophosphate[13]. These starting materials are converted to two central intermediates, first geissoschizine **1** and then, through a series of unknown steps, to Wieland–Gumlich aldehyde **6** (refs. [14,15] and Fig. 1; see Supplementary Fig. 2 for full biosynthetic hypothesis). Wieland–Gumlich aldehyde **6** has been proposed to be converted to strychnine **10** through the incorporation of acetate to form the piperidone moiety, although the mechanism of acetate incorporation and ring cyclization has remained unclear[12,13] (ring G in Fig. 1; see Supplementary Fig. 3 for carbon and ring annotations). Subsequent hydroxylations and methylations of strychnine **10** would yield brucine **15** (ref. [16] and Fig. 1).

To identify strychnine biosynthetic genes, we selected two members of the *Strychnos* genus (family: Loganiaceae), one known producer of strychnine **10**, *S. nux-vomica*[17] and one non-producer, *Strychnos* sp.[18], to investigate this biosynthetic pathway. Metabolic analysis of

*S. nux-vomica* revealed the presence of several strychnos alkaloids, including strychnine **10**, isostrychnine **11**, β-colubrine **13** and brucine **15**, all of which accumulate in the roots (Supplementary Fig. 4). These alkaloids were absent in the non-producer, although a biosynthetically related compound, strychnos alkaloid diaboline **8**, was detected in its roots and stems (Supplementary Fig. 5). We generated tissue-specific RNA-sequencing data from these two plants to enable gene discovery.

The biosynthetic pathway of geissoschizine **1** from tryptophan and geranyl pyrophosphate has been completely elucidated in the phylogenetically related plant *Catharanthus roseus* (family: Apocynaceae) (see Supplementary Fig. 6 for the phylogenetic relationship of *C. roseus* and *S. nux-vomica*). *C. roseus* produces monoterpene indole alkaloids unrelated to strychnine[19]. A homologue for each biosynthetic gene in the geissoschizine **1** pathway was readily identified in the *S. nux-vomica* transcriptome, suggesting that the biosynthetic pathway of geissoschizine **1** is conserved in *C. roseus* and *S. nux-vomica*. These genes are all expressed preferentially in *S. nux-vomica* roots (Supplementary Fig. 7), consistent with previous feeding studies that suggest strychnine **10** biosynthesis occurs primarily in the roots[12,13]. Candidate genes for subsequent steps were selected according to three criteria: (1) high expression in the roots of *S. nux-vomica* (fragments per kilobase of transcript per million mapped reads (FPKM) ≥ 20); (2) co-expression with putative upstream genes; and (3) genes that could encode proteins with catalytic functions that are consistent with the chemical logic of our hypothesized biosynthetic pathway (Fig. 2).

The chemical steps for transformation of geissoschizine **1** to Wieland–Gumlich aldehyde **6** are not known. However, given the structural similarity between the Wieland–Gumlich aldehyde **6** and the known early alkaloid intermediate dehydropreakuammicine **2** (ref. [19] and Fig. 3a), chemical logic suggests that Wieland–Gumlich aldehyde **6** could form from dehydropreakuammicine **2** through ester hydrolysis, decarboxylation, oxidation and reduction (Supplementary Fig. 2). If this hypothesis is correct, *S. nux-vomica* should contain a homologue of geissoschizine oxidase, which has also been isolated from *C. roseus* (*Cr*GO). In vitro, *Cr*GO converts geissoschizine **1** to akuammicine **3**,

[1]Department of Natural Product Biosynthesis, Max-Planck Institute for Chemical Ecology, Jena, Germany. [2]Microscopic Imaging Service Group, Max-Planck Institute for Chemical Ecology, Jena, Germany. ✉e-mail: oconnor@ice.mpg.de

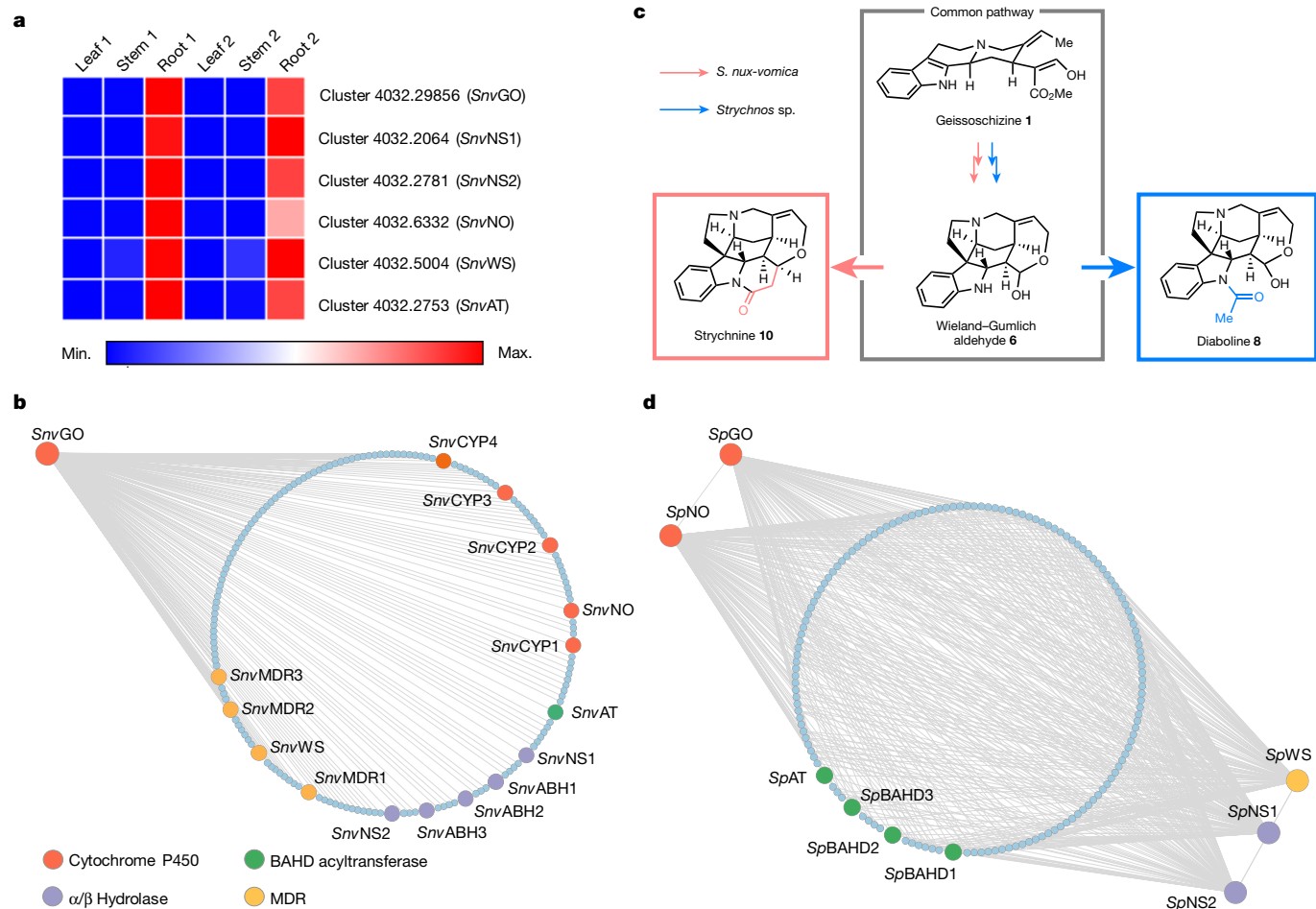

**Fig. 1 | The proposed biosynthesis pathway for strychnine and brucine.** The partial biosynthetic pathway was predicted on the basis of previous radioisotopic feeding experiments. OPP, pyrophosphate; GPP, geranyl pyrophosphate.

presumably through the spontaneous deformylation of dehydro-preakuammicine **2** (ref. [19], Fig. 3a and Supplementary Fig. 2). A BLAST search using *Cr*GO as the query against the *S. nux-vomica* transcriptome identified one hit (transcript cluster 4032.29856; CYP71AY6) with 46% amino-acid sequence identity (Supplementary Fig. 8) that showed similar expression profiles with upstream biosynthetic gene candidates (Fig. 2a). We expressed this gene in *N. benthamiana* leaves through *Agrobacterium tumefaciens*-mediated transient expression followed by infiltration of geissoschizine **1**. Liquid chromatography–mass spectrometry analysis of leaf extracts revealed the deformylation product of dehydropreakuammicine, akuammicine **3** (Fig. 3b and Extended Data Fig. 1). Therefore, cluster 4032.29856 was named *Snv*GO.

Because it is known that decarboxylation of a methyl ester can be triggered by ester hydrolysis[20], we speculated that an α/β hydrolase[20,21] would hydrolyse the ester moiety of dehydropreakuammicine **2** and therefore lead to decarboxylation before spontaneous deformylation to akuammicine **3** occurs. This would result in the formation of the strychnos alkaloid norfluorocurarine **4** (Fig. 3a and Supplementary Fig. 2). On the basis of a co-expression analysis using *Snv*GO as bait, we initially selected five α/β hydrolases (r ≥ 0.95, Pearson correlation coefficient) for functional characterization (Fig. 2b). Each was tested in *N. benthamiana* along with *Snv*GO and geissoschizine **1** as substrate. Two of these candidates (clusters 4032.2064 and 4032.2781) led to the production of norfluorocurarine **4**, along with substantially

**Fig. 2 | Expression analysis of candidate genes in *S. nux-vomica* (strychnine producer) and *Strychnos* sp. (diaboline producer).** Both strychnine and diaboline are derived from the same biosynthetic intermediate, the Wieland–Gumlich aldehyde. **a**, Expression profiles of identified genes in *S. nux-vomica*. The expression of each identified gene is represented as the FPKM of *S. nux-vomica* transcriptomes. Sample sets 1 and 2 represent two biological replicates. **b**, Co-expression analysis using *Snv*GO as bait in *S. nux-vomica*. The circle of dots represents genes co-expressed with *Snv*GO (Pearson's r ≥ 0.95; 470 genes in total). **c**, *S. nux-vomica* and *Strychnos* sp. share a common pathway from geissoszhizine **1** to Wieland–Gumlich aldehyde **6**. **d**, Co-expression analysis in *Strychnos* sp. *Sp*GO, *Sp*NS1, *Sp*NS2, *Sp*NO and *Sp*WS were used as baits. The circle depicts genes co-expressed with all the baits (r > 0.6; 3,999 genes in total). Enlarged and annotated dots in (**b** and **d**) represent genes tested in *N. benthamiana*.

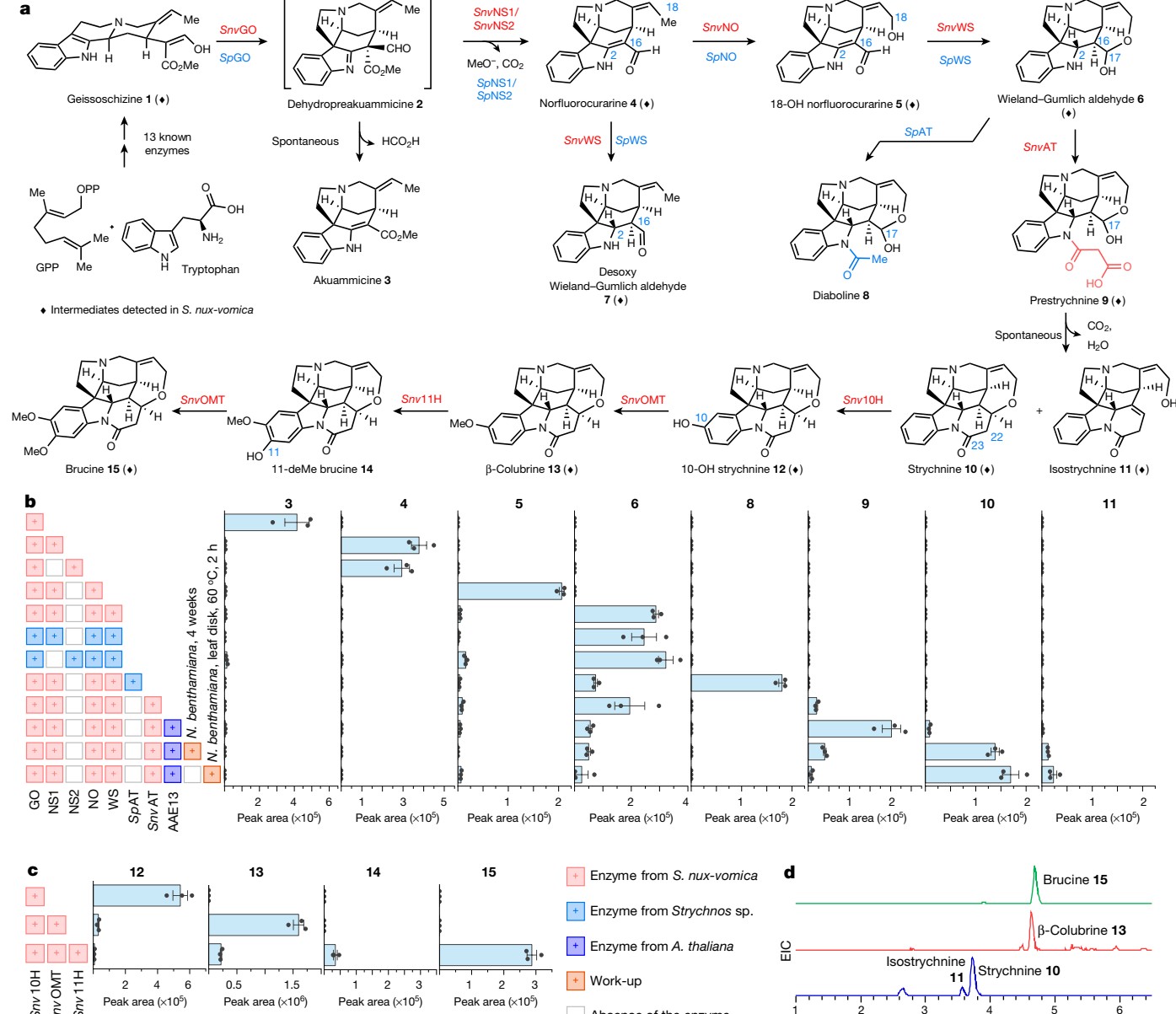

**Fig. 3 | Discovery of a diaboline, strychnine and brucine biosynthesis pathway. a**, The complete biosynthetic pathway leading to the production of diaboline **8**, strychnine **10** and brucine **15**. Diamonds represent intermediates detected in *S. nux-vomica* (Extended Data Fig. 10). **b**, The liquid chromatography–mass spectrometry peak area of products produced in *N. benthamiana* after expression of the indicated enzymes and geissoschizine **1** infiltration. Date are mean ± s.e.m.; *n* = 3 biological replicates. **c**, The liquid chromatography–mass spectrometry peak area of products produced in *N. benthamiana* after

expression of the indicated enzymes and strychnine **10** infiltration. Date are mean ± s.e.m.; *n* = 3 biological replicates. Work-up: manipulation after expression of the indicated enzymes and substrate infiltration. **d**, Extracted ion chromatograms (EIC) for strychnine **10**, isostrychnine **11**, β-colubrine **13** and brucine **15** in *N. benthamiana* leaves expressing all nine enzymes with the infiltration of geissoschizine **1** and disodium malonate. All intermediates were validated by comparison to synthetic authentic standards (see Supplementary Information for synthetic procedures).

decreased levels of the deformylation product akuammicine **3** (Fig. 3b and Extended Data Fig. 1). Therefore, we named these two α/β hydrolases norfluorocurarine synthase 1 and 2 (*Snv*NS1 and *Snv*NS2). *Snv*NS1 and *Snv*NS2 share 74% identity at the protein level and showed the same reactivity in the *N. benthamiana* transient-expression system. We used *Snv*NS1 in all subsequent experiments.

To convert norfluorocurarine **4** to Wieland–Gumlich aldehyde **6**, a hydroxylase and a reductase are required to install the C18 hydroxyl group and reduce the 2,16 double bond, respectively (Fig. 3a). A total of five cytochrome P450 proteins[22] and four medium-chain dehydrogenase/reductases (MDRs)[23] that were co-expressed (*r* ≥ 0.95)

(Fig. 2b) with *Snv*GO were initially considered, because these two protein families are often involved in alkaloid biosynthesis. Because the order of hydroxylation and reduction is unknown, combinatorial transient-expression experiments in *N. benthamiana*[24–26] were adopted. Simultaneous expression of all candidate cytochrome P450 proteins and MDRs in *N. benthamiana* leaves combined with *Snv*GO and *Snv*NS1 indeed resulted in the consumption of norfluorocurarine **4** and production of Wieland–Gumlich aldehyde **6** (Fig. 3a). Co-infiltration of one cytochrome P450 (cluster 4032.6332; CYP71A144) along with *Snv*GO, *Snv*NS1 and geissoschizine **1** in *N. benthamiana* leaves produced a hydroxylated product 18-OH norfluorocurarine **5** that co-eluted with

the synthetic standard (Fig. 3b and Extended Data Fig. 2). Intermediate **5** is consumed after one candidate MDR (cluster 4032.5004) is added to the co-infiltration experiments and the accumulation of Wieland–Gumlich aldehyde **6** is observed (Fig. 3b and Extended Data Fig. 3). Therefore, we named this cytochrome P450 norflurocurarine oxidase (*Snv*NO) and the MDR Wieland–Gumlich aldehyde synthase (*Snv*WS). Notably, in planta and in vitro assays showed that *Snv*WS could reduce the 2,16 double bond in both norflurocurarine **4** and 18-OH norflurocurarine **5** (Fig. 3a, Extended Data Fig. 3 and Supplementary Fig. 9). Stereoselective reduction by *Snv*WS is probably initiated by the tautomerization of the enamine moiety in **4** and **5** through protonation at the α face, followed by NADPH reduction at the β face. The subsequent spontaneous cyclization between the C18-OH and C16 aldehyde, possibly facilitated by the conformational flexibility of the reduced substrate, forms the hemiacetal in **6** (Supplementary Fig. 10). In vitro steady-state kinetics indicated that *Snv*WS had a higher catalytic efficiency with **5** than with **4** ($k_{cat}/K_m = 0.297$ min$^{-1}$ µM$^{-1}$ for **5** compared with 0.068 min$^{-1}$ µM$^{-1}$ for **4**) (Supplementary Fig. 11). A model of *Snv*WS docked with 18-OH norflurocurarine **5** suggests that Thr95 and Ser309 in *Snv*WS may hydrogen bond with the C18 hydroxyl group in 18-OH norflurocurarine **5**, providing an explanation for the differences in catalytic efficiency between norflurocurarine **4** and 18-OH norflurocurarine **5** (Supplementary Fig. 10). No cytochrome P450, including *Snv*NO, could hydroxylate desoxy Wieland–Gumlich aldehyde **7**, suggesting that the order of the reactions is first oxidation to form 18-OH norflurocurarine **5**, followed by reduction.

To complete the biosynthesis of strychnine **10** from Wieland–Gumlich aldehyde **6**, a new piperidone ring containing two additional carbon atoms must be installed (ring G in Fig. 1). However, the intermediates or the reaction steps for this ring construction are not known; the only clue is that the additional two-carbon unit (C22 and C23) originates from [$^{14}$C]acetate[12,13]. To facilitate the discovery of these cryptic late biosynthetic steps, we compared the strychnine producing and non-producing *Strychnos* plants. Metabolic analysis showed that the major alkaloid in the non-strychnine producer *Strychnos* sp. is diaboline **8** (Supplementary Fig. 5), a compound that is most likely derived from *N*-acetylation of Wieland–Gumlich aldehyde **6** (Fig. 3a). Therefore, we hypothesized that *S. nux-vomica* and *Strychnos* sp. should share the same biosynthetic pathway from geissoschizine **1** to Wieland–Gumlich aldehyde **6** (Fig. 2c). Indeed, a BLAST search against the non-producer transcriptome identified orthologues *Sp*GO (CYP71AY7, 92% amino-acid identity to *Snv*GO), *Sp*NS1 (92% amino-acid identity to *Snv*NS1), *Sp*NS2 (88% amino-acid identity to *Snv*NS2), *Sp*NO (CYP71A145, 91% amino-acid identity to *Snv*NO) and *Sp*WS (93% amino-acid identity to *Snv*WS). To validate the function of these genes, we expressed them in two combinations (*Sp*Go, *Sp*NS1, *Sp*NO and *Sp*WS; and *Sp*Go, *Sp*NS2, *Sp*NO and *Sp*WS) in *N. benthamiana* leaves with co-infiltration of geissoschizine **1**. Both combinations led to the formation of Wieland–Gumlich aldehyde **6** (Fig. 3b and Extended Data Fig. 4). The only remaining step for the biosynthesis of diaboline **8** is the acetylation of the indole amine (Fig. 3a), which in alkaloid biosynthesis is often catalysed by a BAHD acyltransferase using acetyl-CoA as an acyl donor[27]. Four BAHD acyltransferase candidates were co-expressed with all five genes (*r* > 0.6) (Fig. 2d). Transient expression of one candidate (*Sp*AT) with upstream genes generated diaboline **8** in *N. benthamiana* (Fig. 3b and Extended Data Fig. 4).

*S. nux-vomica* contains an orthologue (cluster 4032.2753; *Snv*AT) of *Sp*AT (85% amino-acid identity to *Sp*AT) that is highly expressed in the roots and showed high expression correlation with previously identified genes (*r* ≥ 0.99 with each gene) (Fig. 2a,b). However, *S. nux-vomica* does not produce diaboline **8**, and previous feeding studies demonstrated that diaboline **8** is not a biosynthetic precursor of strychnine **10** (ref. [14]). We surmised that *Snv*AT and *Sp*AT may have distinct enzymatic activities, and indeed, simultaneous expression of *Snv*AT and *Snv*GO, *Snv*NS1, *Snv*NO, *Snv*WS and geissoschizine **1** in *N. benthamiana* led

to only trace levels of diaboline **8**. However, a new compound with a mass corresponding to a malonylated product was detected in the leaf extracts, which suggested that *Snv*AT is a BAHD acyltransferase with predominantly malonyltransferase activity (Fig. 3b and Extended Data Fig. 5). Although the expression of this enzyme in *N. benthamiana* resulted in only the partial consumption of Wieland–Gumlich aldehyde **6**, we hypothesized that the conversion might be limited by the low concentration of malonyl-CoA in *N. benthamiana* leaves. Therefore, we expressed these enzymes along with AAE13 (*Arabidopsis thaliana*), a cytosolic enzyme that produces malonyl-CoA accessible to cytosolic *Snv*AT[28] (Supplementary Fig. 12). The addition of AAE13 and co-infiltration of the co-substrate disodium malonate to the transient-expression system resulted in a tenfold increase in the production of malonylated product (Fig. 3b and Extended Data Fig. 5). During purification, this product rapidly decomposed, so we treated the crude methanolic extracts of *N. benthamiana* leaves with trimethylsilyldiazomethane to methylate the carboxylic acid, followed by aldehyde reduction with sodium borohydride. The derivatized products were confirmed by comparison to synthetic standards (Supplementary Fig. 13), indicating that the *Snv*AT product was *N*-malonyl Wieland–Gumlich aldehyde **9** (Fig. 3a). Therefore, although *Snv*AT and *Sp*AT share 85% amino acid identity, they have distinct catalytic activities. Phylogenetic analysis showed that *Snv*AT clusters with *Sp*AT in an acetyltransferase clade, which is evolutionarily distinct from the canonical malonyltransferase clade (Supplementary Fig. 14). Homology models of *Snv*AT and *Sp*AT[29] (Supplementary Fig. 15) were used to identify one amino acid (*Snv*AT(R424F) and *Sp*AT(F421R)) that controls the selectivity between acetyl and malonyl transferase activity (Supplementary Figs. 16 and 17). These models suggest that the arginine residue is responsible for the malonyl-CoA selectivity by forming a bidentate salt bridge with the carboxylate of malonyl-CoA[30,31] (Supplementary Fig. 18), providing a straightforward mechanistic explanation for the difference in alkaloid accumulation in these two plants. Notably, the 17-*O*-acylation product was predominant in in vitro assays at physiological pH (Supplementary Fig. 19), which may be because of changes in the protein activity in a non-cellular environment or differences in the equilibration of the open and closed forms of the Wieland–Gumlich aldehyde substrate.

Notably, a trace amount of strychnine **10** and isostrychnine **11** could be detected in the methanolic extracts of *N. benthamiana* leaves that produce malonylated Wieland–Gumlich aldehyde **9** (Fig. 3b and Extended Data Fig. 6). These two alkaloids accumulated and **9** decreased over time when stored at room temperature (Supplementary Fig. 22). Indeed, most of **9** was converted to strychnine **10** and isostrychnine **11** in *N. benthamiana* leaves that were harvested 4 weeks after infiltrating the substrates (Fig. 3b and Extended Data Fig. 6). Incubating **9** with recombinant *Snv*AT or *N. benthamiana* crude protein extracts did not accelerate the conversion of **9** to **10** (Supplementary Fig. 23). These experiments suggest that conversion of **9** to strychnine **10** and isostrychnine **11** could occur spontaneously both in vitro and under physiological conditions. Alternatively, heating *N. benthamiana* leaves at 60 °C for 2 h substantially accelerated the conversion (Fig. 3b and Extended Data Fig. 6). We think that **10** and **11** are formed through the decarboxylation of the β-keto acid moiety in **9** to form an α,β-unsaturated amide. Subsequent *oxa*-Michael addition by C18 hydroxyl group would generate strychnine **10**. The α,β-unsaturated amide can also tautomerize to the β,γ-unsaturated amide to form isostrychnine **11** (Supplementary Fig. 24).

Previous radioisotopic labelling studies indicated that a structurally uncharacterized biosynthetic intermediate could be converted to strychnine by warming the acid extracts from *S. nux-vomica* roots[14,15]. The reported chemical properties of this intermediate[14,15], which was called prestrychnine (see Supplementary Fig. 2 for the previously proposed structure), are similar to **9**. Therefore, we suggest that the proposed structure of prestrychnine be revised to **9**. Notably, in this feeding

study the levels of radioisotope-labelled prestrychnine was 9 times higher than strychnine **10** after 3 days of feeding of *S. nux-vomica* with [14]C-tryptophan[14], suggesting that the conversion of prestrychnine to strychnine **10** is a slow process in *S. nux-vomica*. Indeed, we screened numerous α/β hydrolases[21,32] and polyketide synthases[33], as well as members of these two families that are known to catalyse decarboxylation of β-keto acid functionalities, and we also screened numerous transporters that could transfer prestrychnine to the vacuole where the acidic environment might accelerate the decarboxylation. However, none of these gene candidates accelerated the formation of strychnine **10** and isostrychnine **11**. To establish whether conversion of prestrychnine to strychnine is a slow, non-enzymatic process in *S. nux-vomica*, we performed hydroponic feeding of deuterium-labelled Wieland–Gumlich aldehyde **6** to the roots of *S. nux-vomica*. Labelled prestrychnine **9** could be detected after 3 days, but trace amounts of strychnine **10** and isostrychnine **11** appeared only after 7 days (Extended Data Fig. 7). Collectively, these data are consistent with the previously published experiments[14,15] and with the rate of strychnine formation in our heterologous expression system. The fact that prestrychnine **9** is converted to strychnine **10** slowly in *S. nux-vomica* is consistent with a non-enzymatic process, although the involvement of an enzyme with only modest rate acceleration cannot be definitively ruled out.

Brucine **15**, which is a dimethoxylated derivative of strychnine **10**, is also highly accumulated in the roots of *S. nux-vomica* (Fig. 3a and Supplementary Fig. 4). To identify the hydroxylase, 12 full-length cytochrome P450 proteins that shared a relatively high co-expression correlation with *Snv*GO (Pearson's *r* > 0.7) were selected for subsequent tests (Supplementary Table 1). When one cytochrome P450 (cluster 4032.17050; CYP82D367) was expressed in the presence of strychnine **10** in *N. benthamiana*, 10-OH strychnine **12** was formed (strychnine-10-hydroxylase (*Snv*10H)) (Fig. 3c and Extended Data Fig. 8). The presence of β-colubrine **13** in *S. nux-vomica* suggests that the two methoxy groups are installed sequentially (Fig. 3a), so we next identified five methyltransferases[34] that were highly expressed in the roots of *S. nux-vomica* (Supplementary Table 2). Expression of one of the methyltransferases (cluster 4032.16453; *Snv*OMT) with *Snv*10H in *N. benthamiana* resulted in the formation of a compound corresponding to synthetic β-colubrine **13** (Fig. 3c and Extended Data Fig. 8). None of the aforementioned 12 co-expressed cytochrome P450 proteins catalysed the hydroxylation of β-colubrine **13**, but the high accumulation of the final product brucine **15** in roots led us to identify all 13 other cytochrome P450 proteins that were strongly expressed (FPKM ≥ 20) in roots (Supplementary Table 1). Of these 13 proteins, we initially targeted the 3 within the CYP71 clade (Supplementary Fig. 25). One of these cytochrome P450 proteins (cluster 4032.16581; CYP71AH44, *Snv*11H)— assayed in combination with strychnine, *Snv*10H and *Snv*OMT—produced brucine **15** as a major product along with trace amounts of the hydroxylated product 11-deMe brucine **14** (Fig. 3c and Extended Data Fig. 9). When we infiltrated synthetic β-colubrine **13** into tobacco leaves that express *Snv*11H alone only 11-deMe brucine **14** is formed; brucine **15** is formed only in the presence of *Snv*OMT (Extended Data Fig. 9). In vitro and in planta assays showed that *Snv*OMT could also methylate 11-OH strychnine **16** to α-colubrine **17** (Supplementary Fig. 26), and 10-deMe brucine **18** to brucine **15** (Supplementary Fig. 27), although with lower efficiency. Overall, these results highlight the promise for production of strychnos-type alkaloids using synthetic biology approaches, although substantial optimization of the heterologous host production system is required.

Having completed the pathway of brucine **15**, we then reconstituted the pathway in *N. benthamiana* from geissoschizine **1**. We transiently expressed all of the enzymes (*Snv*GO, *Snv*NS1, *Snv*NO, *Snv*WS and *Snv*AT, AAE13, *Snv*10H, *Snv*OMT and *Snv*11H) in tobacco leaves followed by infiltrating geissoschizine **1** and disodium malonate. If the tobacco leaves were harvested 1 week after infiltrating the substrates, the accumulation of strychnine **10**, isostrychnine **11**, β-colubrine **13** and brucine **15** was observed (Fig. 3d and Supplementary Fig. 28). Additionally, all of the intermediates in the pathway except for 11-deMe brucine **14** could be detected in the roots of *S. nux-vomica* (Fig. 3a and Extended Data Fig. 10), suggesting that the heterologously reconstituted pathway in *N. benthamiana* matches the physiologically relevant pathway in *Strychnos* plants.

Here we report the discovery of nine enzymes that convert geissoschizine **1** to diaboline **8**, strychnine **10** and brucine **11**, using a combination of chemical logic, -omics datasets and enzymatic characterization. Pioneering studies of the structure and synthesis of strychnine provided the foundation for discovery of the enzymes of strychnine biosynthesis as it occurs in nature. These discoveries not only shed light on how plants produce these diverse alkaloids, but also provide a genetic basis for heterologous production of strychnos alkaloid derivatives to discover potent lead compounds through metabolic engineering approaches, providing a new challenge for synthetic biology.

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

## Reporting summary

Further information on research design is available in the Nature Research Reporting Summary linked to this paper.

## Data availability

The sequence of genes characterized in this article are deposited in the National Center for Biotechnology (NCBI) GenBank under the following accession numbers: *Snv*GO (OM304290), *Snv*NS1 (OM304291), *Snv*NS2 (OM304292), *Snv*NO (OM304293), *Snv*WS (OM304294), *Snv*AT (OM304295), *Snv*10H (OM304296), *Snv*OMT (OM304297), *Snv*11H (OM304298), *Sp*GO (OM304299), *Sp*NS1 (OM304300), *Sp*NS2 (OM304301), *Sp*NO (OM304302), *Sp*WS (OM304303) and *Sp*AT (OM304304). The raw reads from the RNA-sequencing profiling analysis of *S. nux-vomica* and *Strychnos* sp. are deposited in the NCBI Sequence Read Archive (SRA) database under the BioProject accessions PRJNA825510 and PRJNA826736, respectively. Source data are provided with this paper.

**Acknowledgements** We thank S. Arndt and C. Löhne for providing plant material; M. Florean for her help with feeding experiments; D. Ayled Serna Guerrero and M. Kunert for assistance with mass spectrometry; Y. Nakamura and C. Paetz for assistance with NMR analysis; E. Rothe and the greenhouse team for taking care of the plants; D. Nelson for his assistance in the systematic naming of the cytochrome P450 enzymes characterized in this study. This work was supported by grants from the European Research Council (788301) and the Max Planck Society. B.H. is grateful to the Alexander von Humboldt Foundation for a postdoctoral fellowship.

**Author contributions** B.H. and S.E.O'C. designed the study and wrote the manuscript; B.H. performed all synthesis, cloning, protein-expression and protein-assay experiments; D.G. assisted with transcriptome preparation and analysis; L.C. assisted with enzyme cloning and data analysis; P.S. cloned AAE13 from *A. thaliana* cDNA and assisted with transient expression; C.E.R.L. helped with bioinformatic analyses; M.O.K. provided advice with cloning and protein purification; N.J.H.L. carried out protein homology modelling; and V.G. performed microscopy.

**Funding** Open access funding provided by Max Planck Society.

**Competing interests** The authors declare no competing interests.

## Additional information

**Correspondence and requests for materials** should be addressed to Sarah E. O'Connor.

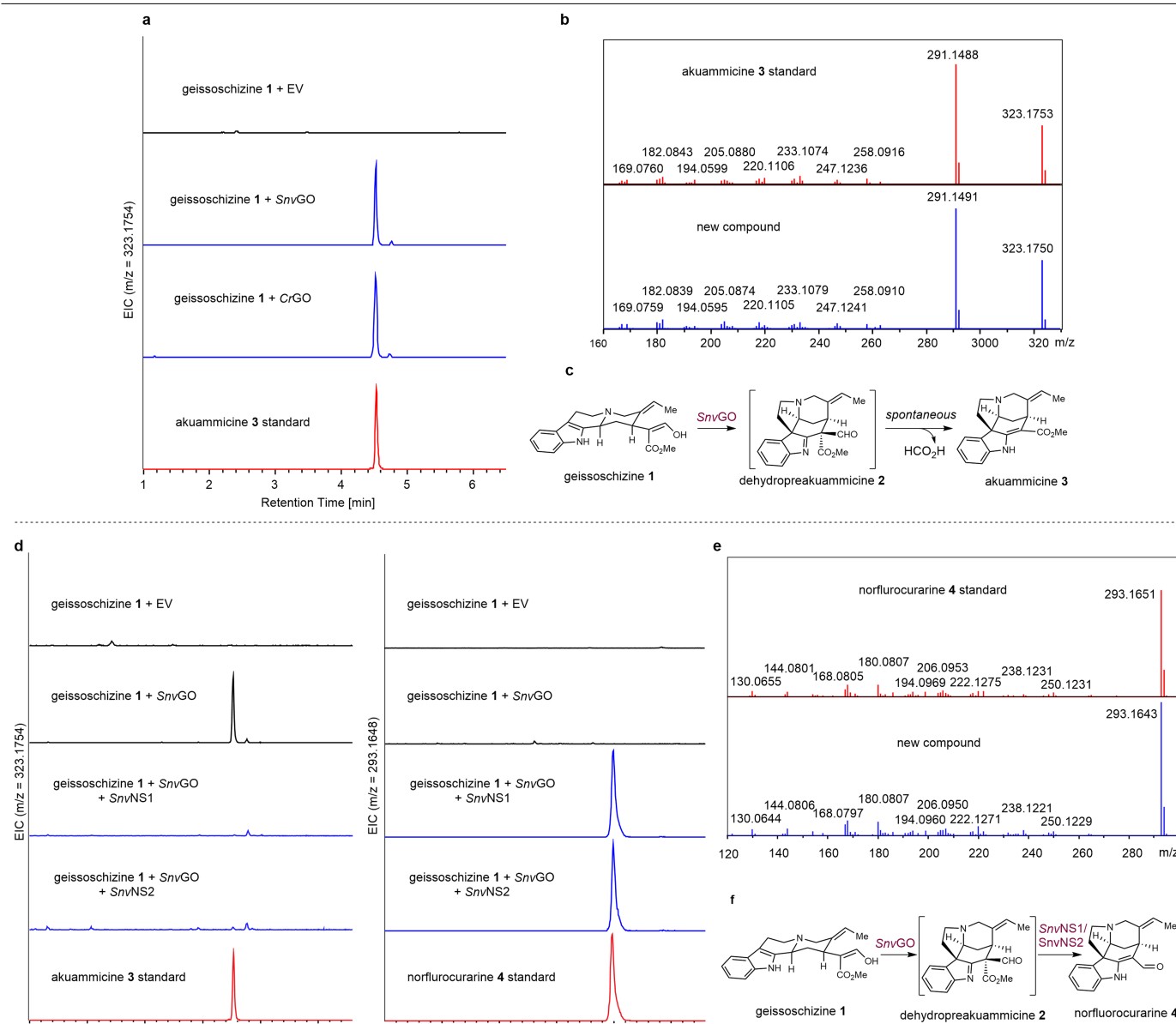

**Extended Data Fig. 1 | Functional characterization of *Snv*GO, *Snv*NS1 and *Snv*NS2. a**. Transient expression of *Snv*GO in *N. benthamiana* with co-infiltration of geissoschizine **1**. Extracted ion chromatograms for akuammicine **3** (*m/z* [M+H]⁺ = 323.1754 ± 0.05). *Cr*GO was used as a positive control. This experiment was repeated three times with similar results. **b**. MS/MS (20 to 50 eV) spectra of akuammicine **3** produced in *N. benthamiana* (blue) compared to standard (red). **c**. Reaction catalyzed by *Snv*GO.

**d**. Transient expression of *Snv*GO, *Snv*NS1, and *Snv*NS2 in *N. benthamiana* with co-infiltration of geissoschizine **1**. Extracted ion chromatograms for akuammicine **3** (*m/z* [M+H]⁺ = 323.1754 ± 0.05, left) and norflurocurarine **4** (*m/z* [M+H]⁺ = 293.1648 ± 0.05, right). This experiment was repeated three times with similar results. **e**. MS/MS (20 to 50 eV) spectra of norflurocurarine **4** produced in *N. benthamiana* (blue) compared to standard (red). **f**. Reaction catalyzed by *Snv*NS1 and *Snv*NS2. EV, empty vector.

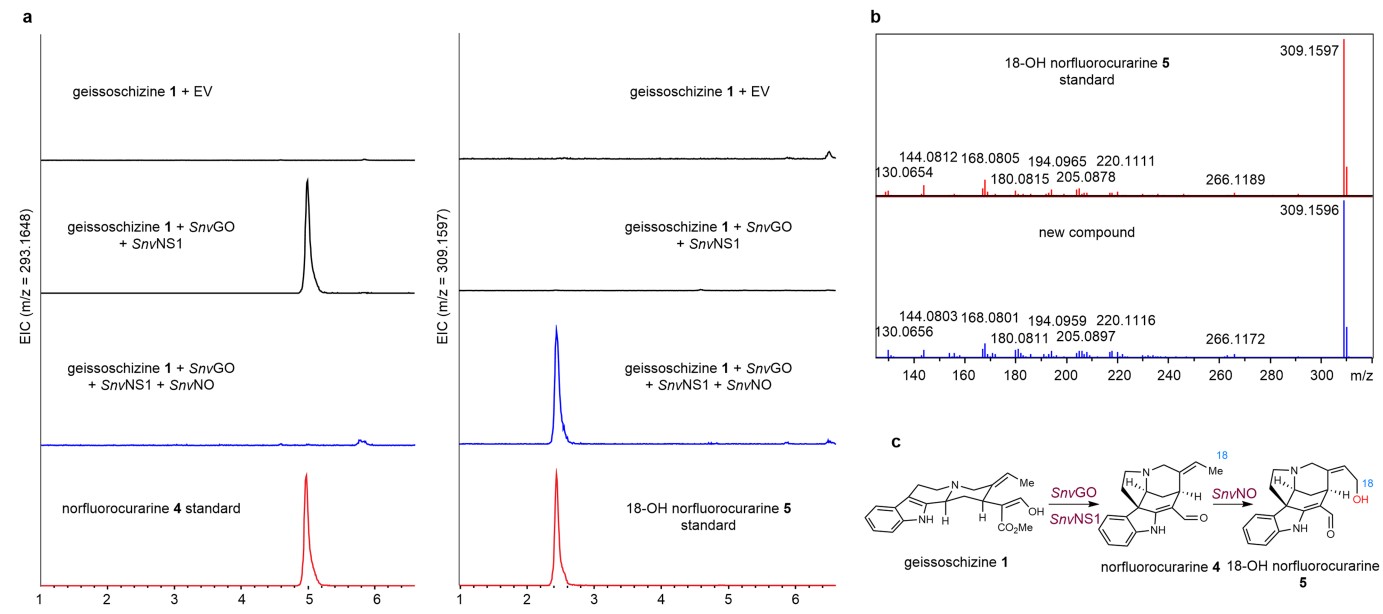

**Extended Data Fig. 2 | Functional characterization of _Snv_NO. a**. Transient expression of _Snv_GO, _Snv_NS1, and _Snv_NO in _N. benthamiana_ with co-infiltration of geissoschizine **1**. Extracted ion chromatograms for norflurocurarine **4** (_m/z_ [M+H]⁺ = 293.1648 ± 0.05, left) and 18-OH norflurocurarine **5** (_m/z_ [M+H]⁺ = 309.1567 ± 0.05, right). This experiment was repeated three times with similar results. **b**. MS/MS (20 to 50 eV) spectra of 18-OH norflurocurarine **5** produced in _N. benthamiana_ (blue) compared to standard (red). **c**. Reaction catalyzed by _Snv_NO.

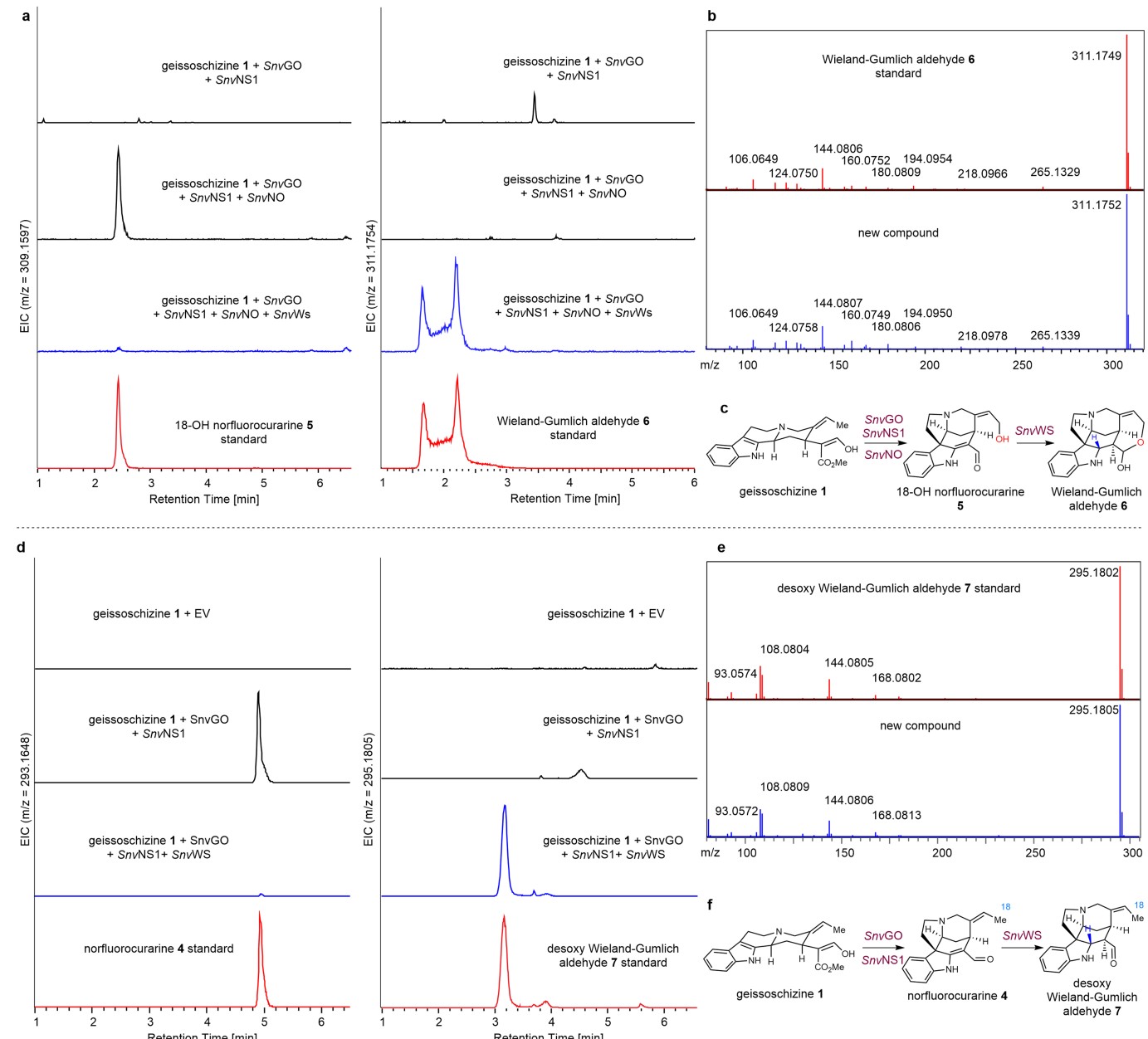

**Extended Data Fig. 3 | Functional characterization of *Snv*WS. a**. Transient expression of *Snv*GO, *Snv*NS1, *Snv*NO and *Snv*WS in *N. benthamiana* with co-infiltration of geissoschizine **1**. Extracted ion chromatograms for 18-OH norflurocurarine **5** ($m/z$ [M+H]$^+$ = 309.1567 ± 0.05, left) and Wieland-Gumlich aldehyde **6** ($m/z$ [M+H]$^+$ = 311.1754 ± 0.05, right). The broad two peaks of Wieland-Gumlich aldehyde **6** in chromatogram due to the hemiacetal diastereomers. This experiment was repeated three times with similar results. **b**. MS/MS (20 to 50 eV) spectra of Wieland-Gumlich aldehyde **6** produced in *N. benthamiana* (blue) compared to synthetic standard (red). **c**. Reaction catalyzed by *Snv*GO, *Snv*NS1, *Snv*NO and *Snv*WS. **d**. Transient expression of *Snv*GO, *Snv*NS1, and *Snv*WS in *N. benthamiana* with co-infiltration of geissoschizine **1**. Extracted ion chromatograms for norflurocurarine **4** ($m/z$ [M+H]$^+$ = 293.1648 ± 0.05, left) and desoxy Wieland-Gumlich aldehyde **7** ($m/z$ [M+H]$^+$ = 295.1805 ± 0.05, right). This experiment was repeated three times with similar results. **e**. MS/MS (20 to 50 eV) spectra of desoxy Wieland-Gumlich aldehyde **7** produced in *N. benthamiana* (blue) compared to synthetic standard (red). **f**. Reaction catalyzed by *Snv*GO, *Snv*NS1, and *Snv*WS.

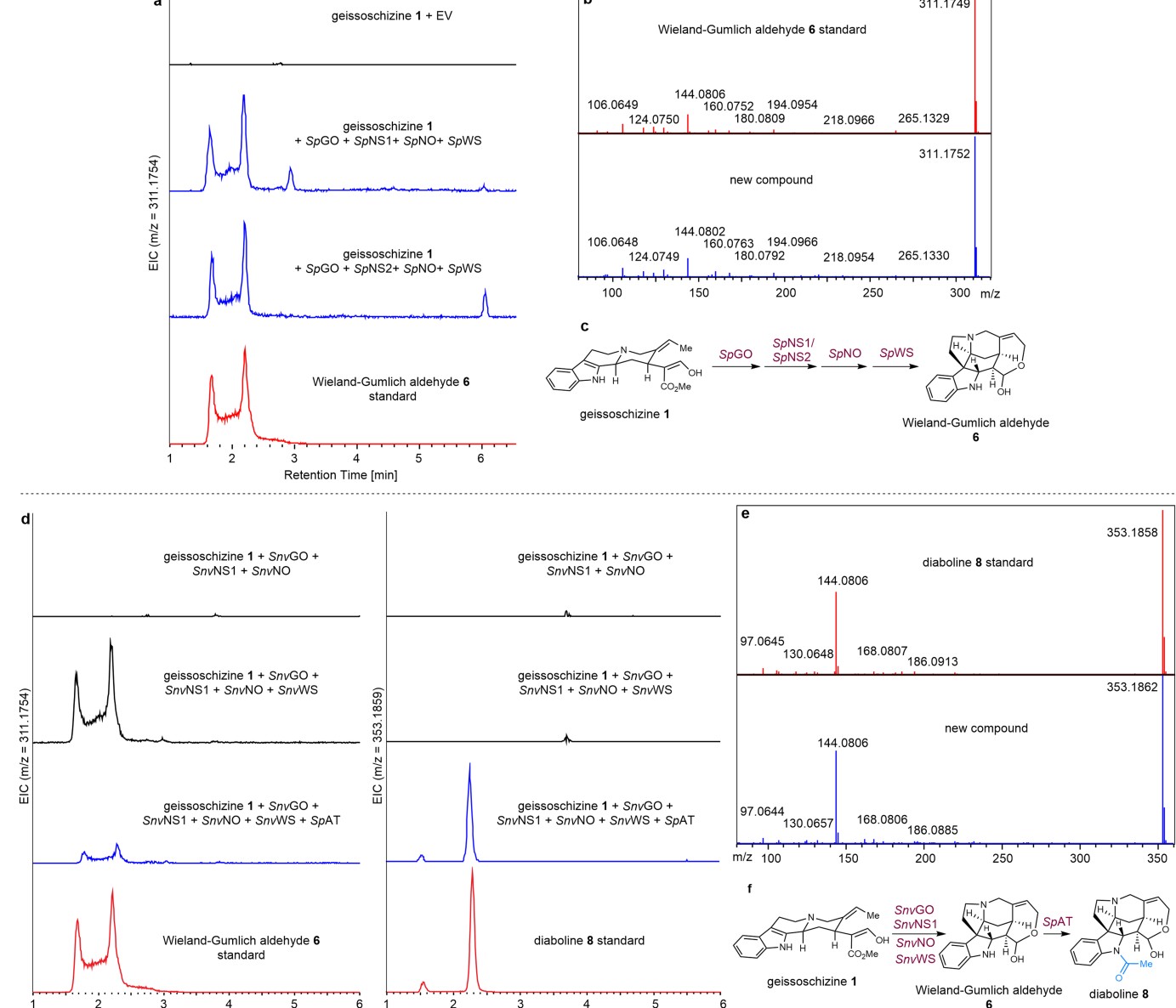

**Extended Data Fig. 4 | Functional characterization of *Sp*GO, *Sp*NS1, *Sp*NS2, *Sp*NO, *Sp*WS and *Sp*AT. a.** Transient expression of *Sp*GO, *Sp*NS1, *Sp*NS2, *Sp*NO, and *Sp*WS in *N. benthamiana* with co-infiltration of geissoschizine **1**. Extracted ion chromatograms for Wieland-Gumlich aldehyde **6** ($m/z$ [M+H]$^+$ = 311.1754 ± 0.05). This experiment was repeated three times with similar results. **b.** MS/MS (20 to 50 eV) spectra of Wieland-Gumlich aldehyde **6** produced in *N. benthamiana* (blue) compared to synthetic standard (red). **c.** Reaction catalyzed by *Sp*GO, *Sp*NS1, *Sp*NS2, *Sp*NO, and *Sp*WS. **d.** Transient expression of *Snv*GO, *Snv*NS1, *Snv*NO,

*Snv*WS and *Sp*AT in *N. benthamiana* with co-infiltration of geissoschizine **1**. Extracted ion chromatograms for Wieland-Gumlich aldehyde **6** ($m/z$ [M+H]$^+$ = 311.1754 ± 0.05, left) and diaboline **8** ($m/z$ [M+H]$^+$ = 353.1859 ± 0.05, right). The two peaks of diaboline **8** in chromatogram due to the hemiacetal diastereomers. This experiment was repeated three times with similar results. **-e.** MS/MS (20 to 50 eV) spectra of diaboline **8** produced in *N. benthamiana* (blue) compared to synthetic standard (red). **f.** Reaction catalyzed by *Sp*AT.

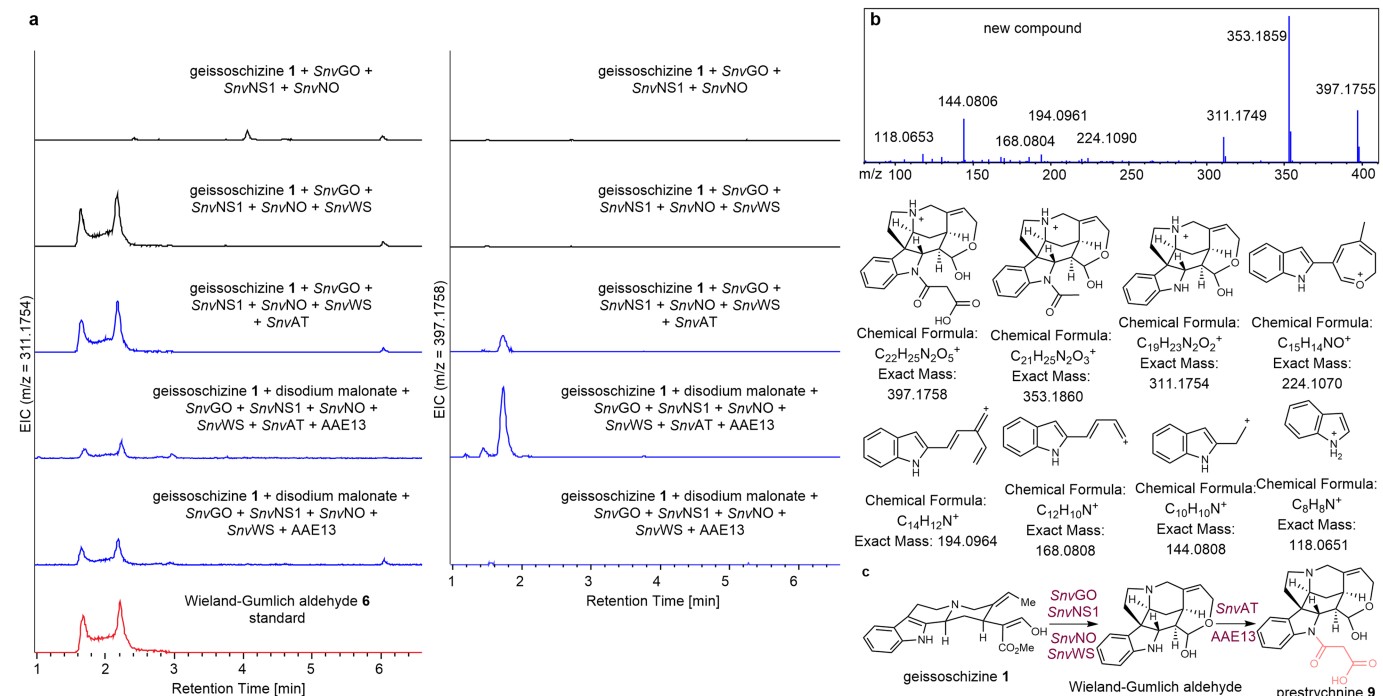

**Extended Data Fig. 5 | Functional characterization of *Snv*AT. a.** Transient expression of *Snv*GO, *Snv*NS1, *Snv*NO, *Snv*WS, *Snv*AT, and AAE13 in *N. benthamiana* with co-infiltration of geissoschizine **1** and disodium malonate. Extracted ion chromatograms for for Wieland-Gumlich aldehyde **6** (*m/z* [M+H]⁺ = 311.1754 ± 0.05, left) and prestrychnine **9** (*m/z* [M+H]⁺ = 397.1758 ± 0.05, right). The two peaks of **9** in chromatogram due to the hemiacetal diastereomers. This experiment was repeated three times with similar results. **b.** MS/MS (20 to 50 eV) spectra and putative ion fragments of generated *m/z* [M+H]⁺ 397.1758. **c.** Reaction catalyzed by *Snv*AT.

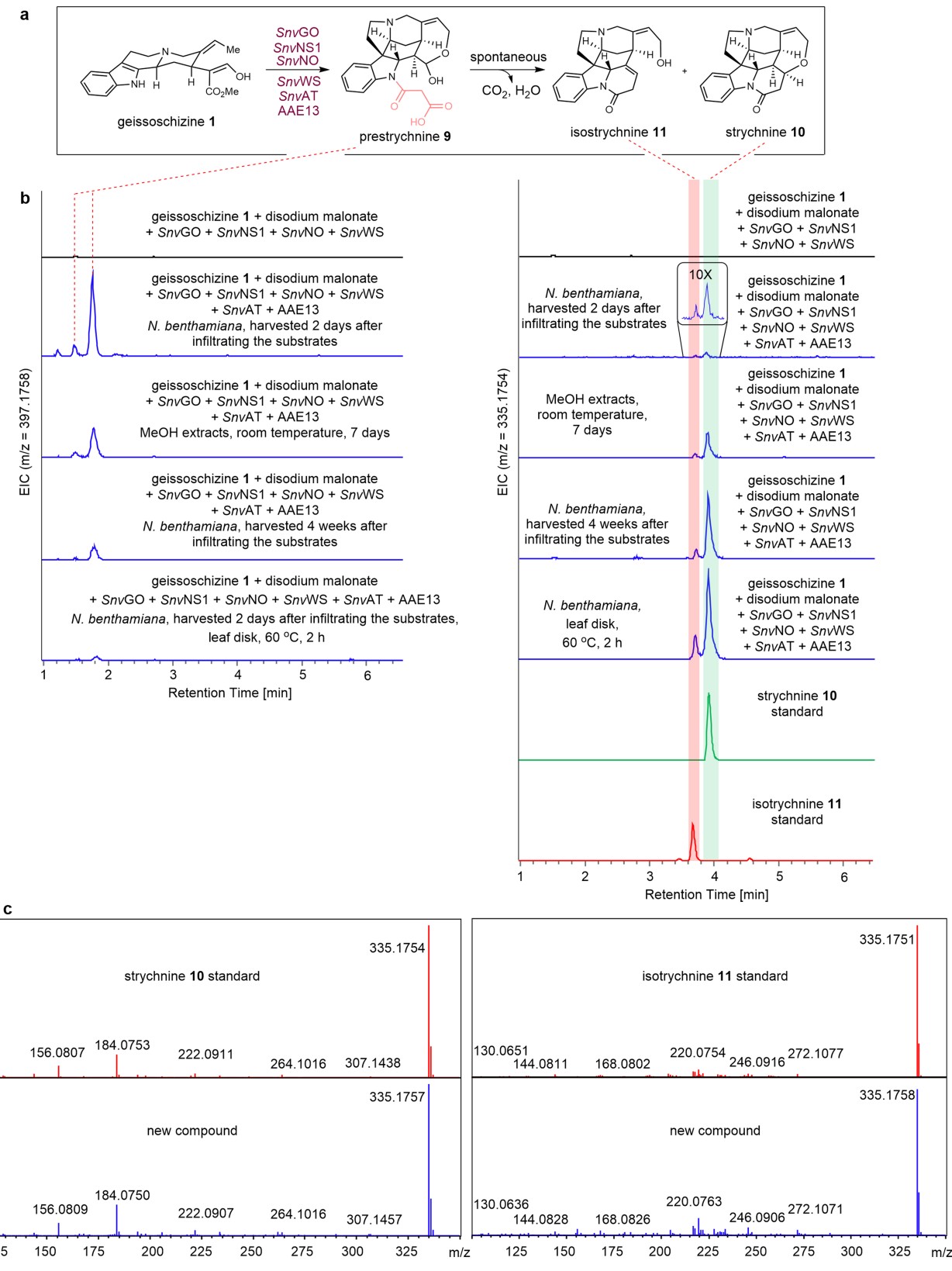

**Extended Data Fig. 6 | Conversion of prestrychnine 9 to strychnine 10 and isostrychnine 11. a.** Reaction of the conversion. **b.** Extracted ion chromatograms for prestrychnine **9** ($m/z$ [M+H]$^+$ = 397.1758 ± 0.05, left) and strychnine **10** and isostrychnine **11** ($m/z$ [M+H]$^+$ = 335.1754 ± 0.05, right). These experiments were repeated three times with similar results. **c.** MS/MS (20 to 50 eV) spectra of generated $m/z$ [M+H]$^+$ 335.1754 (strychnine **10** and isostrychnine **11**, blue) compared to standards (red).

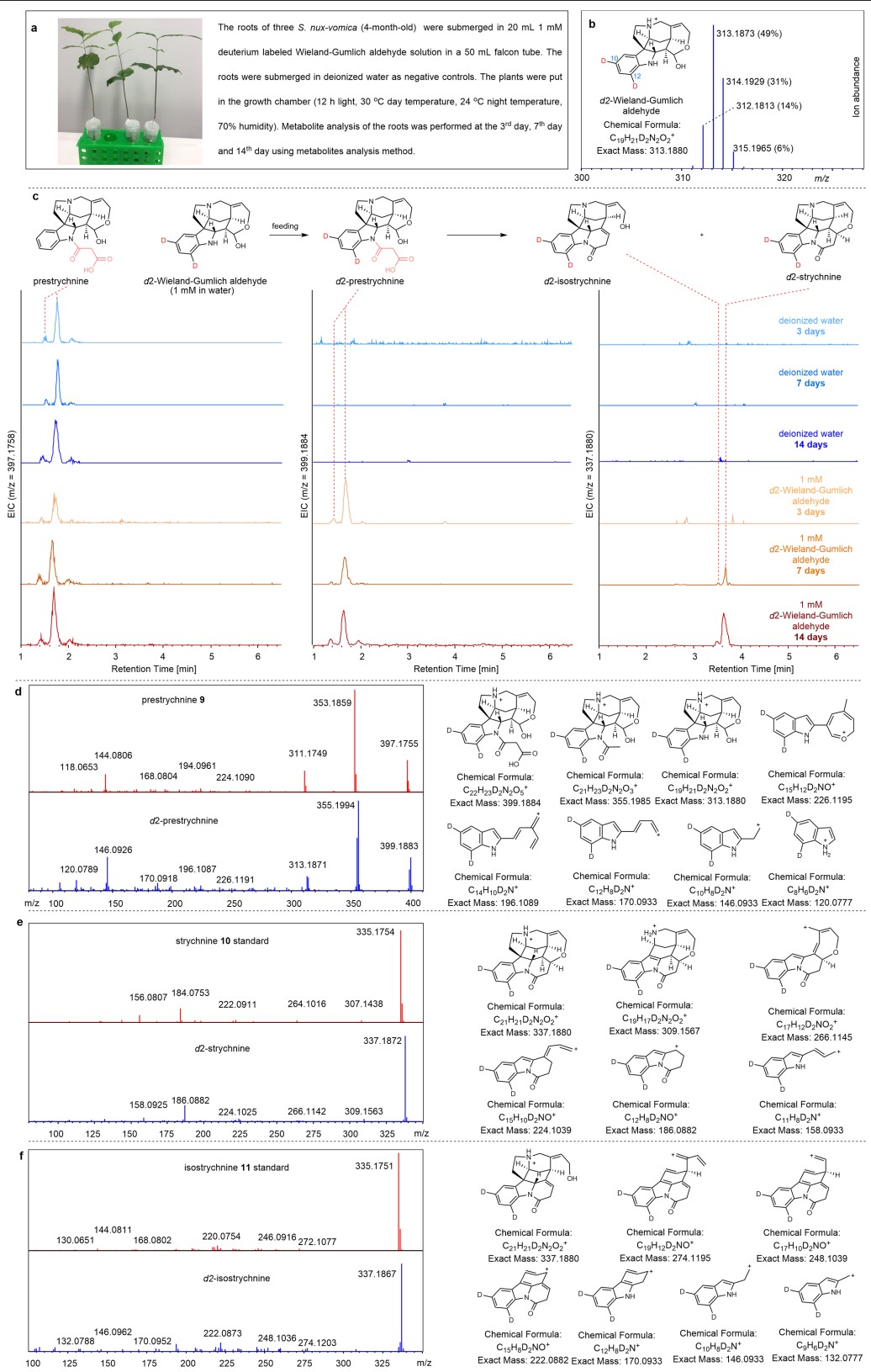

**a** The roots of three *S. nux-vomica* (4-month-old) were submerged in 20 mL 1 mM deuterium labeled Wieland-Gumlich aldehyde solution in a 50 mL falcon tube. The roots were submerged in deionized water as negative controls. The plants were put in the growth chamber (12 h light, 30 °C day temperature, 24 °C night temperature, 70% humidity). Metabolite analysis of the roots was performed at the 3rd day, 7th day and 14th day using metabolites analysis method.

**b** *d2*-Wieland-Gumlich aldehyde

313.1873 (49%)
314.1929 (31%)
312.1813 (14%)
315.1965 (6%)

Chemical Formula: $C_{19}H_{21}D_2N_2O_2^+$
Exact Mass: 313.1880

**c** prestrychnine · *d2*-Wieland-Gumlich aldehyde (1 mM in water) → feeding → *d2*-prestrychnine → *d2*-isostrychnine + *d2*-strychnine

deionized water 3 days
deionized water 7 days
deionized water 14 days
1 mM *d2*-Wieland-Gumlich aldehyde 3 days
1 mM *d2*-Wieland-Gumlich aldehyde 7 days
1 mM *d2*-Wieland-Gumlich aldehyde 14 days

EIC (m/z = 397.1758)
EIC (m/z = 399.1884)
EIC (m/z = 337.1880)

Retention Time [min]

**d** prestrychnine **9**
353.1859
118.0653 144.0806 194.0961 311.1749 397.1755
168.0804 224.1090

*d2*-prestrychnine
355.1994
120.0789 146.0926 196.1087 313.1871 399.1883
170.0918 226.1191

Chemical Formula: $C_{22}H_{23}D_2N_2O_5^+$
Exact Mass: 399.1884

Chemical Formula: $C_{21}H_{23}D_2N_2O_3^+$
Exact Mass: 355.1985

Chemical Formula: $C_{19}H_{21}D_2N_2O_2^+$
Exact Mass: 313.1880

Chemical Formula: $C_{15}H_{12}D_2NO^+$
Exact Mass: 226.1195

Chemical Formula: $C_{14}H_{10}D_2N^+$
Exact Mass: 196.1089

Chemical Formula: $C_{12}H_8D_2N^+$
Exact Mass: 170.0933

Chemical Formula: $C_{10}H_8D_2N^+$
Exact Mass: 146.0933

Chemical Formula: $C_8H_6D_2N^+$
Exact Mass: 120.0777

**e** strychnine **10** standard
335.1754
156.0807 184.0753 222.0911 264.1016 307.1438

*d2*-strychnine
337.1872
158.0925 186.0882 224.1025 266.1142 309.1563

Chemical Formula: $C_{21}H_{21}D_2N_2O_2^+$
Exact Mass: 337.1880

Chemical Formula: $C_{19}H_{17}D_2N_2O_2^+$
Exact Mass: 309.1567

Chemical Formula: $C_{17}H_{12}D_2NO_2^+$
Exact Mass: 266.1145

Chemical Formula: $C_{15}H_{10}D_2NO^+$
Exact Mass: 224.1039

Chemical Formula: $C_{12}H_8D_2NO^+$
Exact Mass: 186.0882

Chemical Formula: $C_{11}H_8D_2N^+$
Exact Mass: 158.0933

**f** isostrychnine **11** standard
335.1751
130.0651 144.0811 168.0802 220.0754 246.0916 272.1077

*d2*-isostrychnine
337.1867
132.0788 146.0962 170.0952 222.0873 248.1036 274.1203

Chemical Formula: $C_{21}H_{21}D_2N_2O_2^+$
Exact Mass: 337.1880

Chemical Formula: $C_{19}H_{12}D_2NO^+$
Exact Mass: 274.1195

Chemical Formula: $C_{17}H_{10}D_2NO^+$
Exact Mass: 248.1039

Chemical Formula: $C_{15}H_8D_2NO^+$
Exact Mass: 222.0882

Chemical Formula: $C_{12}H_8D_2N^+$
Exact Mass: 170.0933

Chemical Formula: $C_{10}H_8D_2N^+$
Exact Mass: 146.0933

Chemical Formula: $C_9H_6D_2N^+$
Exact Mass: 132.0777

**Extended Data Fig. 7** | See next page for caption.

**Extended Data Fig. 7 | Hydroponic feedings to the roots of 4-month-old *S. nux-vomica* seedlings with deuterium labelled Wieland-Gumlich aldehyde. a**. Hydroponic feedings of *S. nux-vomica* in 50 mL falcon tube with 20 mL 1 mM deuterium labelled Wieland-Gumlich aldehyde. **b**. HRMS (ESI) [M+H]$^+$ chromatogram of synthetic deuterium labelled Wieland-Gumlich aldehyde mixture. The major component (49%) is *d2*-Wieland-Gumlich aldehyde. **c**. Extracted ion chromatograms for prestrychnine **9** (*m/z* [M+H]$^+$ = 397.1758 ± 0.005), *d*2-prestrychnine (*m/z* [M+H]$^+$ = 399.1884 ± 0.005), *d*2-isostrychnine (*m/z* [M+H]$^+$ = 337.1880 ± 0.005), *d*2-strychnine (*m/z* [M+H]$^+$ = 337.1880 ± 0.005). **d**. MS/MS (20 to 50 eV) spectrum and putative ion fragments of generated *d*2-prestrychnine (*m/z* [M+H]$^+$ = 399.1884, blue) compared to prestrychnine (*m/z* [M+H]$^+$ = 397.1758, red). **e**. MS/MS (20 to 50 eV) spectra and putative ion fragments of generated *d*2-strychnine (*m/z* [M+H]$^+$ = 337.1880, blue) compared to strychnine standard (*m/z* [M+H]$^+$ = 335.1754, red). **f**. MS/MS (20 to 50 eV) spectra and putative ion fragments of generated *d*2-isostrychnine (*m/z* [M+H]$^+$ = 337.1880, blue) compared to isostrychnine standard (*m/z* [M+H]$^+$ = 335.1754, red). This experiment was repeated three times with similar results.

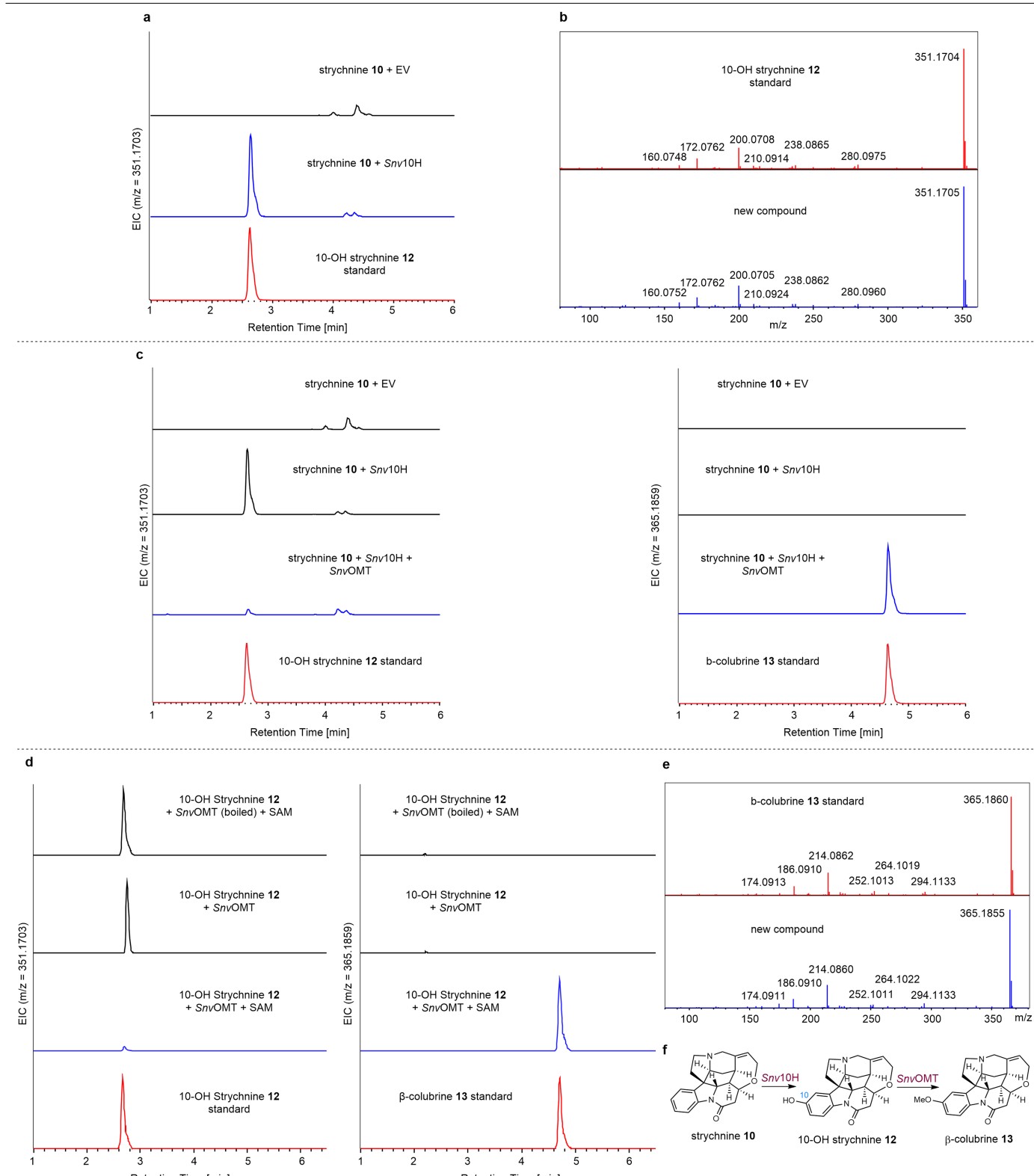

**Extended Data Fig. 8 | Functional characterization of _Snv_10H and _Snv_OMT.**
**a.** Transient expression of _Snv_10H in _N. benthamiana_ with co-infiltration of strychnine **10**. Extracted ion chromatograms for 10-OH strychnine ($m/z$ [M+H]$^+$ = 351.1703 ± 0.05). This experiment was repeated three times with similar results. **b.** MS/MS (20 to 50 eV) spectra of 10-OH strychnine **12** produced in _N. benthamiana_ (blue) compared to standard (red). **c.** Transient expression of _Snv_10H and _Snv_OMT in _N. benthamiana_ with co-infiltration of strychnine **10**. Extracted ion chromatograms for 10-OH strychnine

($m/z$ [M+H]$^+$ = 351.1703 ± 0.05, left) and β-colubrine **13** ($m/z$ [M+H]$^+$ = 365.1859 ± 0.05, right). This experiment was repeated three times with similar results. **d.** _In vitro_ assays using purified _Snv_OMT from SoluBL21 _E. coli_ with 10-OH strychnine **12**. Extracted ion chromatograms for 10-OH strychnine ($m/z$ [M+H]$^+$ = 351.1703 ± 0.05, left) and β-colubrine **13** ($m/z$ [M+H]$^+$ = 365.1859 ± 0.05, right). This experiment was repeated more than three times with similar results. **e.** MS/MS (20 to 50 eV) spectra of β-colubrine **13** produced in _N. benthamiana_ (blue) compared to standard (red). **f.** Reaction catalyzed by _Snv_10H and _Snv_OMT.

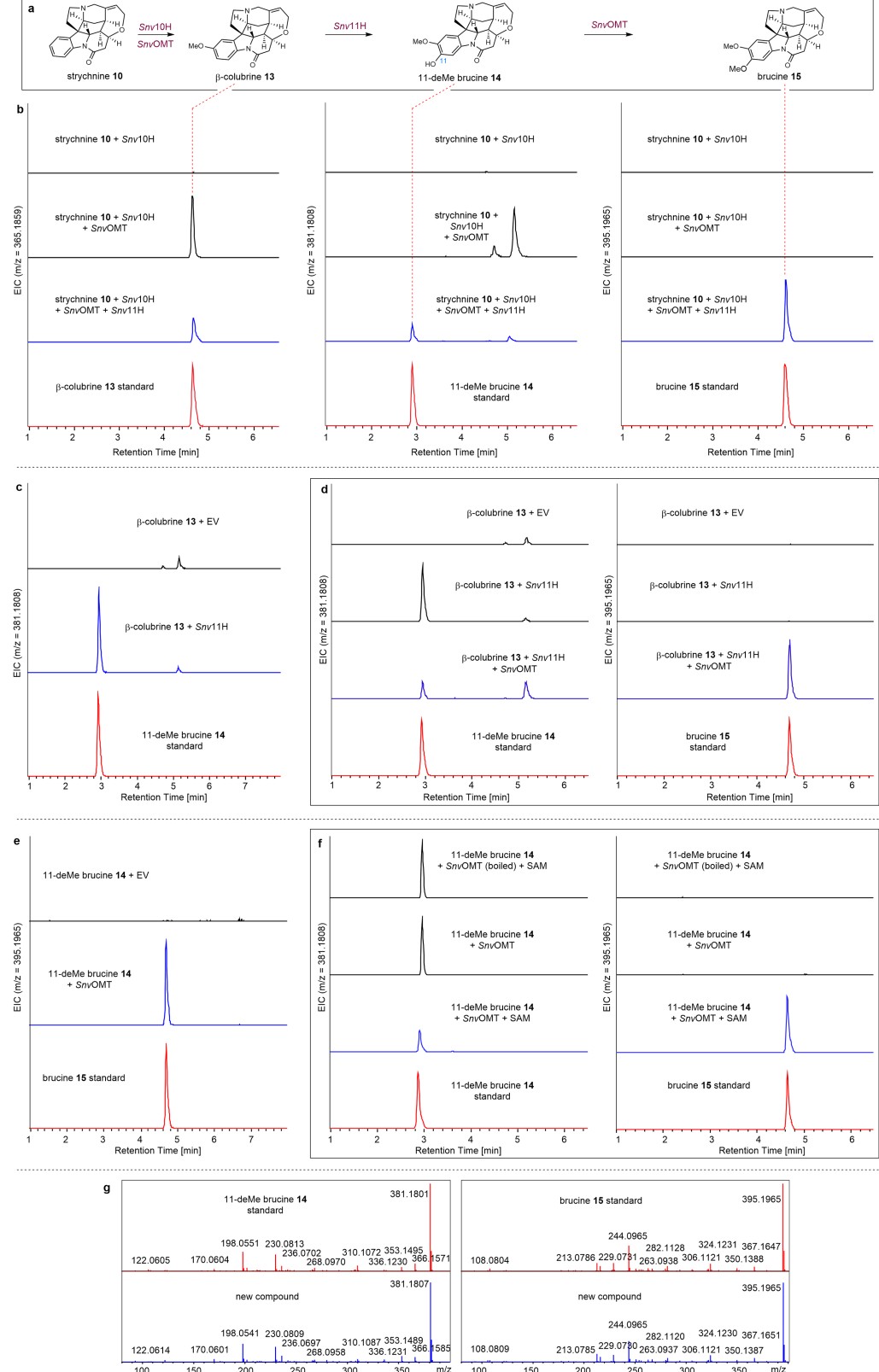

**Extended Data Fig. 9** | See next page for caption.

**Extended Data Fig. 9 | Functional characterization of *Snv*11H. a**. Reaction catalyzed by *Snv*11H and *Snv*OMT. **b**. Transient expression of *Snv*10H, *Snv*OMT, and *Snv*11H in *N. benthamiana* with co-infiltration of strychnine **10**. Extracted ion chromatograms for β-colubrine **13** ($m/z$ [M+H]$^+$ = 365.1859 ± 0.05, left), 11-deMe brucine **14** ($m/z$ [M+H]$^+$ = 381.1808 ± 0.05, middle), and brucine **15** ($m/z$ [M+H]$^+$ = 395.1965 ± 0.05, right). This experiment was repeated three times with similar results. **c**. Transient expression of *Snv*11H in *N. benthamiana* with co-infiltration of β-colubrine **13**. Extracted ion chromatograms for 11-deMe brucine **14** ($m/z$ [M+H]$^+$ = 381.1808 ± 0.05). This experiment was repeated three times with similar results. **d**. Transient expression of *Snv*11H and *Snv*OMT in *N. benthamiana* with co-infiltration of β-colubrine **13**. Extracted ion chromatograms for 11-deMe brucine **14** ($m/z$ [M+H]$^+$ = 381.1808 ± 0.05, left) and brucine **15** ($m/z$ [M+H]$^+$ = 395.1965 ± 0.05, right). This experiment was repeated three times with similar results. **e**. Transient expression of *Snv*OMT in *N. benthamiana* with co-infiltration of 11-deMe brucine **14**. Extracted ion chromatograms for brucine **15** ($m/z$ [M+H]$^+$ = 395.1965 ± 0.05). This experiment was repeated three times with similar results. **f**. *In vitro* assays using purified *Snv*OMT from SoluBL21 *E. coli*. Extracted ion chromatograms for 11-deMe brucine **14** ($m/z$ [M+H]$^+$ = 381.1808 ± 0.05, left) and brucine **15** ($m/z$ [M+H]$^+$ = 395.1965 ± 0.05, right). This experiment was repeated three times with similar results. **g**. MS/MS (20 to 50 eV) spectra of generated 11-deMe brucine **14** and brucine **15** (blue) compared to standards (red).

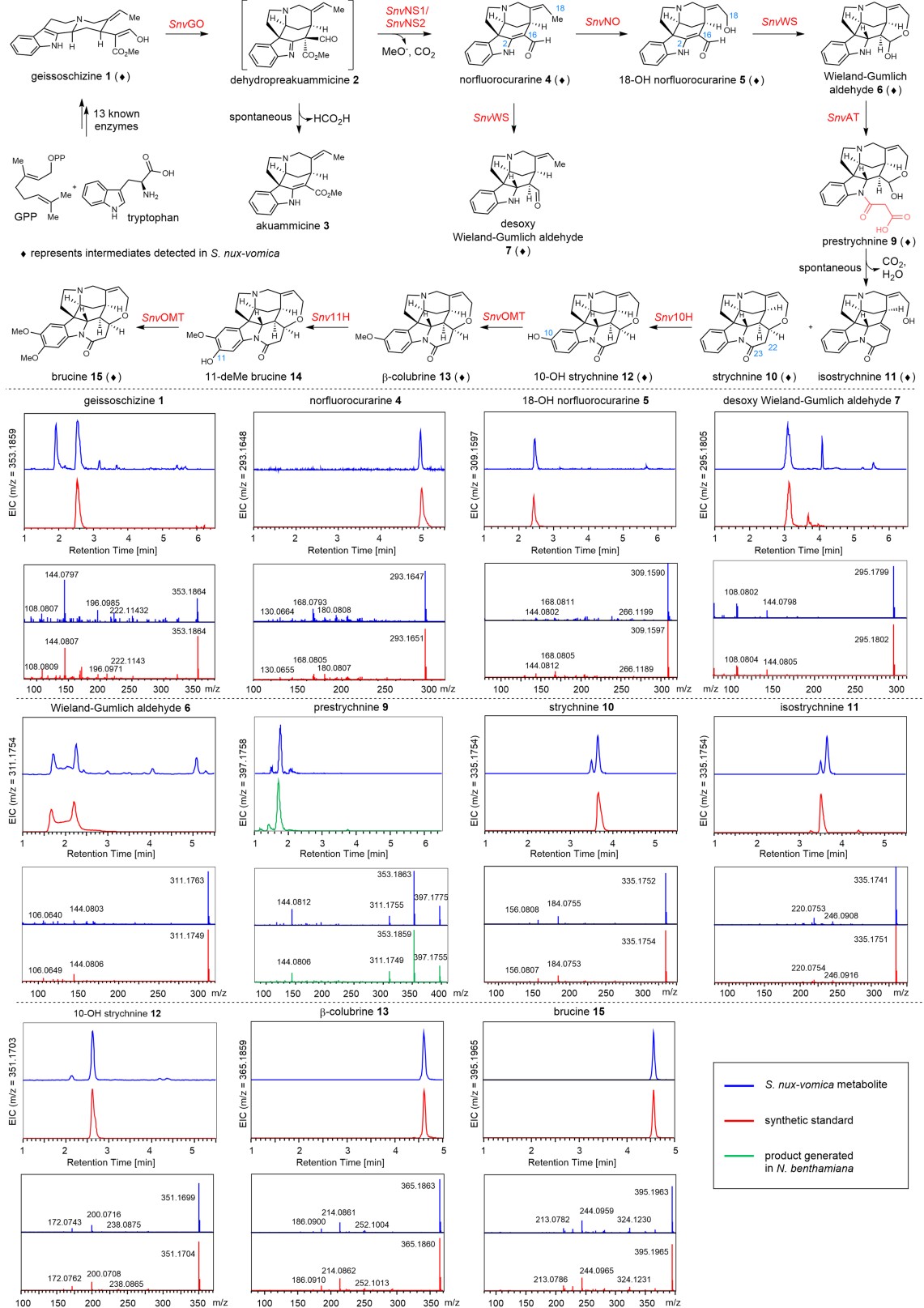

**Extended Data Fig. 10 | Comparison of *S. nux-vomica* metabolites to standards or enzymatic product generated in *N. benthamiana* transient expression.** Intermediates detected in methanolic extracts of *S. nux-vomica* roots (blue) were compared to standards (red) or products produced in *N. benthamiana* transient expression experiment (green) by retention time and MS/MS spectra.

# Reporting Summary

## Statistics

For all statistical analyses, confirm that the following items are present in the figure legend, table legend, main text, or Methods section.

| n/a | Confirmed | |
|---|---|---|
| ☐ | ☒ | The exact sample size (*n*) for each experimental group/condition, given as a discrete number and unit of measurement |
| ☐ | ☒ | A statement on whether measurements were taken from distinct samples or whether the same sample was measured repeatedly |
| ☐ | ☒ | The statistical test(s) used AND whether they are one- or two-sided<br>*Only common tests should be described solely by name; describe more complex techniques in the Methods section.* |
| ☒ | ☐ | A description of all covariates tested |
| ☒ | ☐ | A description of any assumptions or corrections, such as tests of normality and adjustment for multiple comparisons |
| ☐ | ☒ | A full description of the statistical parameters including central tendency (e.g. means) or other basic estimates (e.g. regression coefficient) AND variation (e.g. standard deviation) or associated estimates of uncertainty (e.g. confidence intervals) |
| ☐ | ☒ | For null hypothesis testing, the test statistic (e.g. *F*, *t*, *r*) with confidence intervals, effect sizes, degrees of freedom and *P* value noted<br>*Give P values as exact values whenever suitable.* |
| ☒ | ☐ | For Bayesian analysis, information on the choice of priors and Markov chain Monte Carlo settings |
| ☒ | ☐ | For hierarchical and complex designs, identification of the appropriate level for tests and full reporting of outcomes |
| ☒ | ☐ | Estimates of effect sizes (e.g. Cohen's *d*, Pearson's *r*), indicating how they were calculated |

*Our web collection on statistics for biologists contains articles on many of the points above.*

## Software and code

Policy information about availability of computer code

| Data collection | All presented data have been acquired using existing and routinely used software. LC-MS data was collected by Bruker otofControl 5.2.109/ Hystar 5.1.5.1. NMR data was collected by Bruker TopSpin 3.6.1. Confocal microscopy images were collected by ZEN black 2.1 v.14.0.18.201 (Zeiss, Oberkochen, Germany). |
|---|---|
| Data analysis | The phylogenetic tree was constructed in MEGAX v10.2.0. and visualized with iTOL. Protein homology models were built using the Swiss-Model server and visualized with PyMOL. Molecular docking was performed using AutoDock Vina. The software used for confocal microscopy analysis was ZEN black 2.1 v.14.0.18.201 (Zeiss, Oberkochen, Germany). NMR data were processed with Bruker TopSpin ver. 3.6.1. LC-MS data was processed with Bruker DataAnalysis 5.0 and MetaboScape 4.0. Chemical structures were generated in ChemDraw Professional 17.1. Kinetics data was analysed by GraphPad Prism 8.0.2. Heatmaps were generated by Morpheus: (https://software.broadinstitute.org/ morpheus). Trinity v.2.6.6  was used to perform the transcriptome assembly. CORSET v.4.6 software was used to remove the redundance from Trinity results. Gene expression levels were estimated by RSEM v.1.2.28 and differential expression analysis using DESeq2 v.1.26.0. Coexpression analysis was done using CoExpNetViz software (http://bioinformatics.psb.ugent.be/webtools/coexpr/) and visualized with the Cytoscape v.3.9.0. |

For manuscripts utilizing custom algorithms or software that are central to the research but not yet described in published literature, software must be made available to editors and reviewers. We strongly encourage code deposition in a community repository (e.g. GitHub). See the Nature Portfolio guidelines for submitting code & software for further information.

## Data

Policy information about availability of data

All manuscripts must include a data availability statement. This statement should provide the following information, where applicable:

- Accession codes, unique identifiers, or web links for publicly available datasets
- A description of any restrictions on data availability
- For clinical datasets or third party data, please ensure that the statement adheres to our policy

There are no restrictions on the availability of data. All reported data within this study are available via database or by request. The sequence of genes characterized in this article are deposited in National Center for Biotechnology (NCBI) GenBank under the accession numbers: SnvGO (OM304290), SnvNS1(OM304291), SnvNS2 (OM304292), SnvNO (OM304293), SnvWS (OM304294), SnvAT (OM304295), Snv10H (OM304296), SnvOMT (OM304297), Snv11H (OM304298), SpGO (OM304299), SpNS1 (OM304300), SpNS2 (OM304301), SpNO (OM304302), SpWS (OM304303), SpAT (OM304304). TThe raw reads from the RNA-seq profiling analysis of Strychnos nux-vomica and Strychnos Sp. are deposited in the NCBI Sequence Read Archive (SRA) database under the BioProject accession PRJNA825510 and PRJNA826736, respectively.

# Field-specific reporting

Please select the one below that is the best fit for your research. If you are not sure, read the appropriate sections before making your selection.

☒ Life sciences  ☐ Behavioural & social sciences  ☐ Ecological, evolutionary & environmental sciences

For a reference copy of the document with all sections, see nature.com/documents/nr-reporting-summary-flat.pdf

# Life sciences study design

All studies must disclose on these points even when the disclosure is negative.

| | |
|---|---|
| Sample size | Prior determination of sample size was not a consideration for our data. Replicates of 3 were chosen for heterologous expression experiments in Nicotiana benthamiana. |
| Data exclusions | No data were excluded from the analyses. |
| Replication | The majority of the data presented in this study is representative of three experiments done in different days. All attempts at replication were successful. |
| Randomization | For heterologous expression in Nicotiana benthamiana leaves, each experiment was tested 3 times. Three biological replicates are from different Nicotiana benthamiana plants. Each of these plants would contain one replicate from each different condition. The second pair of fully expanded leaves (counting from the apical meristem side) in each plant were used for experiment. |
| Blinding | Blinding was not relevant for our study. Functional characterization of enzymes or genes required the insight of researchers about the tested samples. |

# Reporting for specific materials, systems and methods

We require information from authors about some types of materials, experimental systems and methods used in many studies. Here, indicate whether each material, system or method listed is relevant to your study. If you are not sure if a list item applies to your research, read the appropriate section before selecting a response.

### Materials & experimental systems

| n/a | Involved in the study |
|---|---|
| ☒ ☐ | Antibodies |
| ☒ ☐ | Eukaryotic cell lines |
| ☒ ☐ | Palaeontology and archaeology |
| ☒ ☐ | Animals and other organisms |
| ☒ ☐ | Human research participants |
| ☒ ☐ | Clinical data |
| ☒ ☐ | Dual use research of concern |

### Methods

| n/a | Involved in the study |
|---|---|
| ☒ ☐ | ChIP-seq |
| ☒ ☐ | Flow cytometry |
| ☒ ☐ | MRI-based neuroimaging |

