## [Peer Review File · Nature]

Manuscript Title: Biosynthesis of strychnine

Reviewer Comments & Author Rebuttals

Reviewer Reports on the Initial Version:

Referees' comments:

Referee #1 (Remarks to the Author):

Strychnine is one of the most well known neurotoxic plant natural products isolated from the *Strychnos nux-vomica* tree. It is a complex polyheterocyclic terpene indole alkaloid, and therefore presents a significant question regarding its biosynthetic mechanism. In this manuscript, Hong et al. used a multi-omics approach to investigate strychnine biosynthesis in *Strychnos nux-vomica*. The authors also used another non-producing *Strychnos* species as control species to assist this effort. Nine enzymes were discovered in this study, namely SnvGO, SnvNS1, SnvNO, SnvWS, SnvAT, AAE13, Snv10H, SnvOMT, and Snv11H. Collectively, these enzymes convert geissoschizine to diaboline, strychnine, and brucine. The authors mostly used *Agrobacterium*-mediated transient expression system in *Nicotiana benthamiana* to demonstrate the activities of the identified enzymes, and were able to reconstitute strychnine biosynthesis in *Nicotiana* by co-infiltration of the pathway genes together with geissoschizine and malonic acid. Overall, it is a very straightforward and well written paper, and solves a historically significant plant natural product biosynthetic problem.

Below are my specific comments:

- 1) The activity of WS is demonstrated in *Nicotiana*. It would be helpful to demonstrate the proposed activity also in vitro using recombinant protein. It is unclear whether WS may also accelerate the hemiacetal formation in 6, in addition to reduction of the 2-16 double bond in 5. The authors may want to model WS and its binding to 5 to provide some insights into the stereochemistry control of the reduction reaction, and discuss its implication on the subsequent hemiacetal formation reaction.
- 2) SnvAT and SpAT with the R424F and F421R substitutions controlling acetyl vs. malonyl transferase activity is one of the highlights of the paper. However, their differential activities were only demonstrated through *Nicotiana* reconstitution experiment. In vitro activities, preferably kinetic properties of these two enzymes, should be shown using recombinant enzymes.
- 3) The authors argue that the conversion of prestrychnine into strychnine is spontaneous and provide various pieces of evidences to support this hypothesis. But these evidences also do not rule out the alternative possible mechanisms that may involve enzyme-catalyzed cyclization in vivo. I suggest the authors tone down their argument on this point and also discuss alternative hypotheses.
- 4) Overall, the manuscript could discuss more on functional implications and uniqueness of each enzyme; and how the impact of this pathway elucidation changes the way we understand the complexity of strychnine biosynthesis and the biomanufacturing process of strychnine.
- 5) The amounts of compounds detected on the MS from *Nicotiana* leaf disks are relatively low; while the discovery of these genes, along with the reconstitution of the biosynthetic pathway is impressive, it is undoubtedly still far from arriving at a feasible engineered system to produce diaboline/strychnine/brucine. Please elaborate on challenges and future efforts to produce enough

would be useful to observe how this change accommodates different acyl-CoA substrates. If possible, it would be useful to include models that include either acetyl-CoA or malonyl-CoA, and the role of this amino acid in the selectivity of these acyl transferases should be discussed.

- The authors show that 9 can slowly be converted into strychnine 10 and, to a lesser extent, isostrychnine 11 over time in *N. benthamiana* without the addition of additional enzymes, and that this can be accelerated with heating. It is reasonably concluded that this reaction may be occurring spontaneously, and slowly, over time, and this is supported by the feeding of labeled 6 to *Strychnos* roots, where fast accumulation of 9 is observed relative to the slower accumulation of 10 and 11. It is therefore claimed that this confirms the formation of 10 and 11 from 9 to be non-enzymatic. While the slower rate of this reaction, and the increased accumulation of 10/11 after heating may support this, additional data would be useful to support the lack of any enzyme involvement (for example, it is possible that low level enzyme activity on 9 could be occurring in *N. benthamiana* and *Strychnos*). Could you provide data for the in vitro enzyme reaction with SnyAT to further confirm this phenomenon when it occurs outside of a living plant system? Some kinetics (even just a time course of product accumulation) for this critical step would really strengthen this hypothesis. This would at least provide evidence against any “low level” enzyme activity in *Strychnos* or *N. benthamiana* that could catalyze the conversion of 9 into 10/11.
- In general, it would be helpful to provide a more rigorous explanation of the expression analysis used to identify candidate pathway genes. E.g., in Figure 2A, what values are illustrated in the heat map? Also, it would be more accurate to state that the heat map illustrates gene expression (relative FPKM values?) rather than describing this panel as “heat map analysis”. I also find the circles in panels B and D not especially informative (vs a list of genes with Pearson’s *r* values) – how many genes are contained in each? The authors describe RNA sequencing of both *Strychnos nux-vomica* and a second strychnine non-producing plant *Strychnos* sp. (is the species unknown?); however, it seems most pathway genes are found in both species (Figure 3). It would be helpful to clarify the putative differing pathways (if that is the authors conclusion) between these two plants in the text. Lastly, given the role of the previously described *Catharanthus* pathway to gessoschizine in the prediction of the strychnine pathway in *Strychnos*, it would be helpful to note the phylogenetic relationship of these two species and highlight the likelihood that the pathways are conserved rather than a result of convergent evolution.

Minor items

- The naming of each identified metabolic enzyme is appropriate. However, I strongly encourage systematic nomenclature of the identified cytochromes P450 (CYPs) so that the phylogenetic lineage and nature of these CYPs is clear and objective.
- It is stated in the Abstract that you reconstitute biosynthesis of strychnine, etc. This isn’t strictly true - consider specifying that this is being accomplished with an infiltrated precursor, not through metabolic engineering of a full pathway.
- For Figures S19 and S20, there is a panel “c” indicated in the figure caption that is not present in the figure itself.
- For Figures S23 and S24, I suggest that the chromatograms shown in each panel are all on the same y-axis. I understand that the 10x zoom is to demonstrate that some level of product is made even without mutation, but it is not immediately clear that one of the mutations leads to a drastic shift in product profile, while the others do not. Perhaps show the chromatograms all on the same

“zoomed-out” scale, then shown a sub-panel that is zoomed in. This is not a necessary item, but may help readers to more easily interpret the data.

- More generally, it would be helpful to either a) label all chromatogram axes, or b) indicate in the figure caption when chromatograms are not on the same scale. There are a few chromatograms where it looks like the scaling is different, and this is not indicated (if the scaling is all the same within a figure, then it would be helpful to state this in the caption for clarity). It looks like this is usually done to emphasize the absence or low abundance of a compound, but that can look misleading at first glance.
- In Fig S25, panel A - it looks like one of your labels shown be “harvested 4 weeks after...”. You repeat “harvested 2 days after...”
- In Figure S29, it is difficult to tell the difference between blue and teal.
- In Figure S38, could you label the peak as “9” - I think that would be helpful for readers.

Author Rebuttals to Initial Comments:

11 April 2022

To the Editor,

We thank both of the reviewers for their detailed and constructive comments, and we have tried to address each one of these comments as completely as possible. We think that the manuscript has been substantially improved as a result of these comments. The revised manuscript highlights in yellow the changes made. A point-by-point response is below.

Referee #1:

Strychnine is one of the most well known neurotoxic plant natural products isolated from the Strychnos nux-vomica tree. It is a complex polyheterocyclic terpene indole alkaloid, and therefore presents a significant question regarding its biosynthetic mechanism. In this manuscript, Hong et al. used a multi-omics approach to investigate strychnine biosynthesis in Strychnos nux-vomica. The authors also used another non-producing Strychnos species as control species to assist this effort. Nine enzymes were discovered in this study, namely SnvGO, SnvNS1, SnvNO, SnvWS, SnvAT, AAE13, Snv10H, SnvOMT, and Snv11H. Collectively, these enzymes convert geissoschizine to diaboline, strychnine, and brucine. The authors mostly used Agrobacterium-mediated transient expression system in Nicotiana benthamiana to demonstrate the activities of the identified enzymes, and were able to reconstitute strychnine biosynthesis in Nicotiana by co-infiltration of the pathway genes together with geissoschizine and malonic acid. Overall, it is a very straightforward and well written paper, and solves a historically significant plant natural product biosynthetic problem.

1) The activity of WS is demonstrated in Nicotiana. It would be helpful to demonstrate the proposed activity also in vitro using recombinant protein. It is unclear whether WS may also accelerate the hemiacetal formation in 6, in addition to reduction of the 2-16 double bond in 5. The authors may want to model WS and its binding to 5 to provide some insights into the stereochemistry control of the reduction reaction, and discuss its implication on the subsequent hemiacetal formation reaction.

Response: We have performed *in vitro* assays using purified SnvWS expressed from SoluBL21 *E. coli*. In these *in vitro* assays, SnvWS could reduce the 2,16 double bond in both norfluorocurarine **4** and 18-OH norfluorocurarine **5** (see Supplementary Fig. 9 in the revised manuscript), which are consistent with the *in vivo* results. Steady state kinetic analyses indicate that the catalytic efficiency for the reduction is 4.3 fold higher for 18-OH norfluorocurarine **5** than it is for **4** (see Supplementary Fig. 11 in the revised manuscript).

We have modeled 18-OH norfluorocurarine **5** in SnvWS using the previously reported crystal structure of *C. roseus* heteroyohimbine synthase THAS2 (63% identity) as a basis for a homology model (see Supplementary Fig. 10 in the revised manuscript). Based on this model, a mechanistic hypothesis for SnvWS is shown below (See Supplementary Fig. 10 in revised manuscript). Briefly, the enamine moiety tautomerizes to the iminium ion to install the C16 chiral center first, and then the iminium ion is reduced by NADPH from the β face to install the C2 chiral center.

The reviewer touches on the important point of hemiacetal formation. The open form of Wieland-Gumlich aldehyde **6** is never observed in our NMR experiments. This is consistent with previous chemical synthesis results (<https://doi.org/10.1021/ja00073a057>, <https://doi.org/10.1021/ja00071a025>), when the C18 hydroxyl group and C16 β aldehyde group is ready, the hemiacetal will be formed spontaneously in the absence of enzyme. Therefore, it is not possible to measure whether *SnvWS* accelerates the already extremely fast spontaneous process of the hemiacetal formation. However, it seems likely that given the increased conformational flexibility afforded by the reduction, the spontaneous formation of the hemiacetal in **6** is greatly favored after action of *SnvWS*.

We do not want to make too many speculations regarding the stereoselective reduction and hemiacetal formation without much more detailed mechanistic experiments that we feel go beyond the scope of this study, but we include some discussion of this issue in the revised manuscript:

“The stereoselective reduction by *SnvWS* is probably initiated by the tautomerization of the enamine moiety in **4** and **5** via protonation at the α face, followed by NADPH reduction at the β face. A subsequent spontaneous cyclization between the C18-OH and C16 aldehyde, possibly facilitated by the conformational flexibility of the reduced substrate, forms the hemiacetal in **6** (Supplementary Fig. 10). *In vitro* steady state kinetics indicated that *SnvWS* showed higher catalytic efficiency with **5** compared to **4** ($K_{cat}/K_m = 0.297 \text{ min}^{-1} \mu\text{M}^{-1}$ for **5** versus $0.068 \text{ min}^{-1} \mu\text{M}^{-1}$ for **4**) (Supplementary Fig. 11). A model of *SnvWS* docked with 18-OH norfluorocurarine **5** suggests that Thr95 and Ser309 in *SnvWS* may hydrogen bond with the C18 hydroxyl group in 18-OH norfluorocurarine **5**, providing an explanation for the different catalytic efficiency between norfluorocurarine **4** and 18-OH norfluorocurarine **5** (Supplementary Fig. 10).”

2) *SnvAT* and *SpAT* with the R424F and F421R substitutions controlling acetyl vs. malonyl transferase activity is one of the highlights of the paper. However, their differential activities were only demonstrated through *Nicotiana* reconstitution experiment. *In vitro* activities, preferably kinetic properties of these two enzymes, should be shown using recombinant enzymes.

Response: When we performed the *in vitro* assays with recombinant enzymes from SoluBL21 *E. coli* or *Nicotiana benthamiana* at physiological pH 7.5 under various conditions (different buffers, cofactor concentrations, different temperatures, coupled with *SnvWS*), we found the major product is the 17-O-malonyl or 17-O-acetyl product, not the N-acylated product as observed *in planta* (Supplementary Fig. 19 in the revised manuscript, which is shown below).

Therefore, we performed an extensive array of *in vitro* assays under different conditions, including a range of different pH values (5.5 to 9.5). At altered pH values, these enzymes showed *in vitro* activity that is consistent with the *in vivo* activity. *SpAT* showed the best N-acetyl selectivity at higher pH (pH 9.5 optimum, Supplementary Fig. 20 in the revised manuscript), while *SnvAT* showed the best N-malonyl selectivity at lower pH (pH 5.5 optimum, Supplementary Fig. 21 in the revised manuscript), while both *SnvAT* and *SpAT* showed low efficiency at optimal pH. This is noted in the manuscript:

“Notably, the O-acylation product was predominant in *in vitro* assays at physiological pH (Supplementary Fig. 19), which may be due to changes in the protein activity in a non-cellular environment, or due to differences in the equilibration of open and closed form of the Wieland-Gumlich aldehyde substrate.”

SpAT also showed robust 17-O malonyl reactivity when malonyl-CoA was used as an acyl donor at pH 5.5 to 8.5 (Supplementary Fig. 20 in the revised manuscript), and *SnvAT* also showed 17-O acetyl reactivity when acetyl-CoA was used as an acyl donor at 5.5 to 8.5 (Supplementary Fig. 21 in the revised manuscript).

We speculate that these selectivity issues may be due to a suboptimal reaction environment for the purified proteins, as Wieland-Gumlich aldehyde **6** was selectively converted to diabolone **8** (N-acetyl selectivity product) or prestrychnine **9** (N-malonyl selectivity product) in *N. benthamiana* transient expression experiments when co-infiltrating *SnvAT* or *SpAT* with Wieland-Gumlich aldehyde **6** (Supplementary Fig. 19 in the revised manuscript, which is shown below). We also speculate that the equilibration between the closed hemiacetal form and the open aldehyde form of the Wieland-Gumlich aldehyde plays a substantial role in this selectivity. This may also be a reason why the *in vivo* experiments are more robust. We speculate that when the pathway enzymes are acting together *in vivo*, there may be more effective shuttling of the open form of the Wieland-Gumlich aldehyde to *SnvAT* or *SpAT*. We hesitate to include these speculative statements in the main text, which require much more extensive experiments, but have added some text as a “Note” to the Supplementary Fig. 19.

“The differences between *in vitro* selectivities under different conditions compared with the robust *in vivo* selectivities is likely due to a suboptimal reaction environment for the purified proteins compared to a cellular environment. We also speculate that the equilibration between the closed hemiacetal form and the open aldehyde form of the Wieland-Gumlich aldehyde plays a substantial role in this selectivity. NMR analysis indicates that the open form of the Wieland-Gumlich aldehyde is not observed *in vitro*, and the availability of the open form of the Wieland-Gumlich aldehyde may be more accessible when this substrate is produced in the cellular environment.”

The detailed mechanism of this process is a fascinating one, and one that we plan to study further. This is an outstanding system in which to measure whether protein-protein interactions are taking place, which are experiments that we are trying to get started at the current time. However, we hope that the reviewer agrees that this initial report of the biosynthetic pathway stands on its own.

Due to this *in vitro* behavior of these two enzymes, we did not test the mutants' reactivity *in vitro*, only in *N. benthamiana*.

3) The authors argue that the conversion of prestrychnine into strychnine is spontaneous and provide various pieces of evidences to support this hypothesis. But these evidences also do not rule out the alternative possible mechanisms that may involve enzyme-catalyzed cyclization in vivo. I suggest the authors tone down their argument on this point and also discuss alternative hypotheses.

Response: We agree with the reviewer that our data do not rule out the possibility that enzyme may involve in the strychnine formation. We have revised this sentence to say:

“The fact that prestrychnine **9** is converted to strychnine **10** slowly in *S. nux-vomica*, is consistent with a non-enzymatic process, though the involvement of an enzyme with only modest rate acceleration cannot be definitively ruled out.”

See also response to Reviewer 2.

4) Overall, the manuscript could discuss more on functional implications and uniqueness of each enzyme; and how the impact of this pathway elucidation changes the way we understand the complexity of strychnine biosynthesis and the biomanufacturing process of strychnine.

Response: We completely agree with this comment, and have attempted to briefly discuss the enzymes throughout the manuscript. Unfortunately, we are at the space limit, so we could not really discuss this in depth. We hope that this can be discussed more extensively in an upcoming review of monoterpene indole alkaloids. Please see below the text that has been incorporated (some of which is also in response to other reviewer comments).

“Since it is known that decarboxylation of a methyl ester can be triggered by ester hydrolysis,²⁰ we speculated that an α/β hydrolase^{20,21} would hydrolyze the ester moiety of dehydropreakuammicine **2**,

“5 cytochromes P450²² and 4 medium-chain dehydrogenase/reductases (MDR)²³ that co-expressed ($r \geq 0.95$, Fig. 2b) with *SnvGO* were considered initially, as these two protein families are widely involved in alkaloid biosynthesis.”

“The stereoselective reduction by *SnvWS* is probably initiated by the tautomerization of the enamine moiety in **4** and **5** via protonation at the α face, followed by NADPH reduction at the β face. A subsequent spontaneous cyclization between the C18-OH and C16 aldehyde, possibly facilitated by the conformational flexibility of the reduced substrate, forms the hemiacetal in **6** (Supplementary Fig. 10). *In vitro* steady state kinetics indicated that *SnvWS* showed higher catalytic efficiency with **5** compared to **4** ($K_{cat}/K_m = 0.297 \text{ min}^{-1} \mu\text{M}^{-1}$ for **5** versus $0.068 \text{ min}^{-1} \mu\text{M}^{-1}$ for **4**) (Supplementary Fig. 11). A model of *SnvWS* docked with 18-OH norfluorocurarine **5** suggests that Thr95 and Ser309 in *SnvWS* may hydrogen bond with the C18 hydroxyl group in 18-OH norfluorocurarine **5**, providing an explanation for the different catalytic efficiency between norfluorocurarine **4** and 18-OH norfluorocurarine **5** (Supplementary Fig. 10).”

“The only remaining step for the diaboline **8** biosynthesis is acetylation of the indole amine (Fig. 3A), which in alkaloid biosynthesis is often catalyzed by a BAHD acyltransferase using acetyl-CoA as an acyl donor.²⁷”

“These models suggest that the arginine residue is responsible for the malonyl-CoA selectivity by forming a bidentate salt bridge with the carboxylate of malonyl-CoA (Supplementary Fig. 18),^{30,31}”

5) The amounts of compounds detected on the MS from Nicotiana leaf disks are relatively low; while the discovery of these genes, along with the reconstitution of the biosynthetic pathway is impressive, it is undoubtedly still far from arriving at a feasible engineered system to produce diaboline/strychnine/brucine. Please elaborate on challenges and future efforts to produce enough quantities in order to compete with the native plant producer.

Response: We agree that the challenges of improving production via synthetic biology/metabolic engineering strategies should be pointed out. We anticipate that in order to optimize production for commercial use, we would have to compare and contrast the promise of different heterologous hosts (i.e. yeast vs *N. benthamiana*), optimize the expression levels of each gene, look at the effects of intracellular compartmentalization, and explore the titers of de novo reconstitution compared to biosynthesis from a synthetic precursor (i.e. geissoschizine). We do not have the space in this format to give these issues justice, but we now note that the production system must be improved. The manuscript now reads:

“Overall, these results highlight the promise for production of strychnos type alkaloids using synthetic biology approaches, though substantial optimization of the heterologous host production system is required.”

6) Please draw the leaving group of the NS reaction in Figure 3A.

Response: Thank you for pointing this out. We added the leaving groups (MeO⁻, CO₂) of the NS reaction in the arrow in Figure 3a.

Referee #2:

In this manuscript, Hong et al. describe the characterization of 9 enzymes that establish the genetic/enzymatic basis for the biosynthesis of the monoterpene indole alkaloid (MIA) strychnine 10, as well as several other structurally-related alkaloids (e.g. brucine, beta-columbine, diaboline) produced in Strychnos species. Strychnine 10 is hypothesized to be derived from a common MIA precursor, geissoschizine 1, and the authors use a previously characterized MIA biosynthetic gene (geissoschizine oxidase, GO) to identify a putative homolog in Strychnos, which they demonstrate to also oxidize geissoschizine 1. The authors then use this identified gene as a “bait” in transcriptional co-expression analysis to identify the remaining biosynthetic genes in strychnine biosynthesis, largely guided by previous studies that had identified likely precursors. The enzymes were all characterized via transient expression in Nicotiana benthamiana, and several were further evaluated in vitro using heterologously produced, purified enzyme. Among these enzymes, the authors identified and characterized an N-malonylating acyl transferase that shares significant homology with an N-acetyltransferase in a Strychnos species that does not produce strychnine. The authors show that strychnine 10 is likely derived from “prestrychnine” 9, the N-malonylated intermediate, via a spontaneous, potentially non-enzymatic transformation, which is supported by substrate feeding with live plants. On top of this, the authors identify a key residue switch between the acyl transferase homologs that is critical for determining whether the enzyme adds a malonyl or acetyl group to the indole amine, thereby determining whether strychnine 10 or diaboline 8 is ultimately synthesized. Finally, the authors identify two oxidases and a methyltransferase that allow for strychnine to be converted into brucine as a “final” product of this pathway. Overall, the content and quality of this manuscript are excellent. The authors have rigorously assessed the metabolism of strychnine and related alkaloids in strychnine-producing and non-producing Strychnos plants, and have elaborated a metabolic route by which these alkaloids can be produced. The characterization of each enzyme is convincing, and supported by either authentic standards, or in one case (prestrychnine 9), robust structural/chemical analysis in the absence of a pure standard. The authors also do a good job of placing this work within the context of the prior, extensive research on strychnine biosynthesis. In my opinion, this work does not require any major additional experiments, but can be strengthened or clarified with some of the following items:

- *It is shown that while the identified MDR (S_{nv}WS) can act on either 4 or 5, the cytochrome P450 responsible for the 18-hydroxylation could not act on 7. This suggests that the order of events is crucial. Because an in vitro reaction system with S_{nv}WS has already been established, measuring the kinetics of 4 vs. 5 as substrates may be useful for helping to support the proposed major order of events.*

Response: We have performed *in vitro* steady state kinetics of S_{nv}WS. S_{nv}WS showed higher catalytic efficiency with **5** than **4** ($K_{cat}/K_m = 0.297 \text{ min}^{-1} \mu\text{M}^{-1}$ for **5** versus $0.068 \text{ min}^{-1} \mu\text{M}^{-1}$ for **4**), which supports the proposed order of the pathway. As suggested by reviewer 1, a model of S_{nv}WS docked with 18-OH norfluorocurarine **5** was created. It predicts that Thr95 and Ser309 in S_{nv}WS likely form hydrogen

bonds with the C18 hydroxyl group in 18-OH norfluorocurarine **5**, which provides an explanation for the different catalytic efficiency between norfluorocurarine **4** and 18-OH norfluorocurarine **5**.

We discuss this in the revised manuscript as below:

“*In vitro* steady state kinetics indicated that *SnvWS* showed higher catalytic efficiency with **5** compared to **4** ($K_{cat}/K_m = 0.297 \text{ min}^{-1} \mu\text{M}^{-1}$ for **5** versus $0.068 \text{ min}^{-1} \mu\text{M}^{-1}$ for **4**) (Supplementary Fig. 11). A model of *SnvWS* docked with 18-OH norfluorocurarine **5** suggests that Thr95 and Ser309 in *SnvWS* may hydrogen bond with the C18 hydroxyl group in 18-OH norfluorocurarine **5**, providing an explanation for the different catalytic efficiency between norfluorocurarine **4** and 18-OH norfluorocurarine **5** (Supplementary Fig. 10).”

- *Through site-directed mutagenesis, the authors show that altering one putative active site residue in SpAT to the corresponding amino acid in SnvAT allow them to switch activity from N-acetylation to N-malonylation of the Wieland-Gumlich aldehyde (6), and vice versa (F421R in SpAt or R424F in SnvAT). The authors present homology models of these proteins to demonstrate the positioning of these key residues in the active site, and while it is clear that this amino acid substitution is crucial, it would be useful to observe how this change accommodates different acyl-CoA substrates. If possible, it would be useful to include models that include either acetyl-CoA or malonyl-CoA, and the role of this amino acid in the selectivity of these acyl transferases should be discussed.*

Response: This is a good suggestion. We have modeled the substrate (Wieland-Gumlich aldehyde **6**) and cofactor (malonyl-CoA) in *SnvAT* (see Supplementary Fig. 18). The arginine residue (R424) is probably responsible for the malonyl-CoA binding via forming a bidentate salt bridge with the carboxylate of malonyl-CoA. This hypothesis is consistent with previous results of protein structures, for example malonyltransferase from *Nicotiana tabacum* (*NtMaT1*, Planta 2012, 236, 781–793, <https://doi.org/10.1007/s00425-012-1660-8>), malonyl-CoA-acyl carrier protein transacylase from *E. coli* (FabD, Acta Cryst. 2006. D62, 613–618, <https://doi.org/10.1107/S0907444906009474>), which are references 30 and 31 in the revised manuscript.

We discussed the role of this amino acid in the selectivity in the manuscript as below:

“Homology models of *SnvAT* and *SpAT* (Supplementary Fig.15)²⁹ were used to identify one amino acid (*SnvAT*-R424F and *SpAT*- F421R) that controls the selectivity between acetyl and malonyl transferase activity (Supplementary Fig. 16, Fig. 17). These models suggest that the arginine residue is responsible for the malonyl-CoA selectivity by forming a bidentate salt bridge with the carboxylate of malonyl-CoA (Supplementary Fig. 18),^{30,31} providing a straightforward mechanistic explanation for the difference in alkaloid accumulation in these two plants.”

- *The authors show that 9 can slowly be converted into strychnine 10 and, to a lesser extent, isostrychnine 11 over time in N. benthamiana without the addition of additional enzymes, and that this can be accelerated with heating. It is reasonably concluded that this reaction may be occurring spontaneously, and slowly, over time, and this is supported by the feeding of labeled 6 to Strychnos roots, where fast accumulation of 9 is observed relative to the slower accumulation of 10 and 11. It is therefore claimed that this confirms the formation of 10 and 11 from 9 to be non-enzymatic. While the slower rate of this reaction, and the increased accumulation of 10/11 after heating may support this, additional data would be useful to support the lack of any enzyme involvement (for example, it is possible that low level enzyme activity on 9 could be occurring in N. benthamiana and Strychnos).*

Could you provide data for the *in vitro* enzyme reaction with *SnvAT* to further confirm this phenomenon when it occurs outside of a living plant system? Some kinetics (even just a time course of product accumulation) for this critical step would really strengthen this hypothesis. This would at least provide evidence against any “low level” enzyme activity in *Strychnos* or *N. benthamiana* that could catalyze the conversion of **9** into **10/11**.

Response: We agree with the reviewer that more evidence to strengthen this hypothesis would be welcome. First, we added a time course (1 day, 7 days, 2 weeks and 4 weeks) of **10/11** accumulation when the methanolic extracts were stored at room temperature (Supplementary Fig. 22). However, we are not sure how much additional mechanistic information this provides for the reviewer.

Please see the response to Reviewer 1, point 2 for a discussion on the *in vitro* activity of *SnvAT*. We thought that a straightforward way to address this reviewer’s point was to use crude plant extracts to see whether they accelerated the *in vitro* conversion of prestrychnine to strychnine. If there is “low level” enzyme activity in *Strychnos* or *N. benthamiana*, then adding protein extracts from these plants to prestrychnine should accelerate the formation of strychnine. We used protein extracts of our heterologous host, *N. benthamiana*, because it is difficult to remove all background strychnos metabolites from *S. nux-vomica* crude protein extracts. Due to the instability of prestrychnine **9** during the purification, we used the crude extracts from *N. benthamiana* leaves which produce prestrychnine **9** to do *in vitro* assays. We also performed the *in vitro* assays with crude protein extracts from *N. benthamiana* along with recombinant *SnvAT*. We tried the assays in two conditions, physiological pH (HEPES buffer pH 7.5, 50 mM) and 50 mM MES buffer pH 5.5 (in which *SnvAT* shows the best N-malonyl selectivity that is observed *in vivo*). After 24 hours, none of these conditions appeared to accelerate the formation of strychnine **10** (the data are provided in Supplementary Fig. 23). Of course, this is a negative result and must be interpreted with caution, and we agree with the reviewer that our current data *does not completely rule out the possibility that strychnine formation may involve an enzyme*. We now say in the manuscript: “The fact that prestrychnine **9** is converted to strychnine **10** slowly in *S. nux-vomica* is consistent with a non-enzymatic process, though the involvement of an enzyme with only modest rate acceleration cannot be definitively ruled out.”

• *In general, it would be helpful to provide a more rigorous explanation of the expression analysis used to identify candidate pathway genes. E.g., in Figure 2A, what values are illustrated in the heat map? Also, it would be more accurate to state that the heat map illustrates gene expression (relative FPKM values?) rather than describing this panel as “heat map analysis”. I also find the circles in panels B and D not especially informative (vs a list of genes with Pearson’s r values) – how many genes are contained in each? The authors describe RNA sequencing of both *Strychnos nux-vomica* and a second strychnine non-producing plant *Strychnos sp.* (is the species unknown?); however, it seems most pathway genes are found in both species (Figure 3). It would be helpful to clarify the putative differing pathways (if that is the authors conclusion) between these two plants in the text.*

Response: We agree that this could be explained better. We have revised the caption of Figure 2a to “Expression profiles of identified genes in *S. nux-vomica*. The expression of each identified gene is represented as the fragments per kilobase of transcript per million mapped reads (FPKM) from *S. nux-vomica* transcriptomes. Sample sets 1 and 2 represent two biological replicates.” It is mentioned in the text that diaboline is made directly from the Wieland-Gumlich aldehyde, which is also a precursor for strychnine. We also re-iterated in the caption of Figure 2 this fact: “Expression analysis of candidate genes in *S. nux-vomica* (strychnine producer) and *Strychnos sp.* (diaboline producer). Both strychnine and diaboline are derived from the same biosynthetic intermediate, the Wieland-Gumlich aldehyde.”

For Figure 2b and 2d, the genes contained in the circles are added in the caption.

Strychnos sp. is likely *Strychnos potatorum* given the chemical profile but since it is not 100% certain, we prefer to be safe and refer to it as simply *Strychnos*.

• *Lastly, given the role of the previously described Catharanthus pathway to geissoschizine in the prediction of the strychnine pathway in Strychnos, it would be helpful to note the phylogenetic relationship of these two species and highlight the likelihood that the pathways are conserved rather than a result of convergent evolution.*

Response: We agree that this should be mentioned. *Catharanthus roseus* is a genus in the family of *Apocynaceae*. The *strychnos* are a genus of plants in the family of *Loganiaceae*. Both *Apocynaceae* and *Loganiaceae* are in the order of *Gentianales*. A phylogenetic tree of *Catharanthus* and *Strychnos* was added in the Supplementary Information (Supplementary Fig. 6 in the revised manuscript).

We added this information in the manuscript as:

“The biosynthetic pathway of geissoschizine **1** from tryptophan and geranyl pyrophosphate (GPP) has been completely elucidated in the phylogenetically related *Apocynaceae* family plant *Catharanthus roseus* (See supplementary Fig. 6 for phylogenetic relationship of *C. roseus* and *S. nux-vomica*), which produces monoterpene indole alkaloids unrelated to strychnine.¹⁹ A homolog for each biosynthetic gene in the geissoschizine **1** pathway was readily identified in the *S. nux-vomica* transcriptome, suggesting that the biosynthetic pathway of geissoschizine **1** is conserved in *C. roseus* and *S. nux-vomica*.”

Minor items

• *The naming of each identified metabolic enzyme is appropriate. However, I strongly encourage systematic nomenclature of the identified cytochromes P450 (CYPs) so that the phylogenetic lineage and nature of these CYPs is clear and objective.*

Response: This is a good suggestion. Professor David R. Nelson (University of Tennessee) helped us to name all the identified cytochromes P450 systematically. We added the information in the manuscript and phylogenetic tree (Supplementary Fig. 8 and 25 in the revised manuscript) in the Supplementary Information. Dr. Nelson is now mentioned in the acknowledgements.

• *It is stated in the Abstract that you reconstitute biosynthesis of strychnine, etc. This isn't strictly true - consider specifying that this is being accomplished with an infiltrated precursor, not through metabolic engineering of a full pathway.*

Response: We thank the reviewer for the suggestion. The abstract has been overall revised to conform with Nature formatting guidelines and we have incorporated this suggestion as below.

“Moreover, we successfully reconstitute strychnine, brucine and diaboline biosynthesis in *Nicotiana benthamiana* from an upstream intermediate...”

• *For Figures S19 and S20, there is a panel “c” indicated in the figure caption that is not present in the figure itself.*

Response: Thank you for pointing this out. We added the missing panels in the revised figure (now panels a and d in Supplementary Fig. 13 in our revised manuscript).

• *For Figures S23 and S24, I suggest that the chromatograms shown in each panel are all on the same y-axis. I understand that the 10x zoom is to demonstrate that some level of product is made even without mutation, but it is not immediately clear that one of the mutations leads to a drastic shift in product profile, while the others do not. Perhaps show the chromatograms all on the same “zoomed-out” scale, then shown a sub-panel that is zoomed in. This is not a necessary item, but may help readers to more easily interpret the data.*

Response: We thank the reviewer for this suggestion. The chromatograms (now in supplementary Fig. 16 and 17 in our revised manuscript) are on the same scale now, and a sub-panel that is zoomed in is shown.

• *More generally, it would be helpful to either a) label all chromatogram axes, or b) indicate in the figure caption when chromatograms are not on the same scale. There are a few chromatograms where it looks like the scaling is different, and this is not indicated (if the scaling is all the same within a figure, then it would be helpful to state this in the caption for clarity). It looks like this is usually done to emphasize the absence or low abundance of a compound, but that can look misleading at first glance.*

Response: We thank the reviewer for this suggestion. Chromatograms in each Figure are on the same scale now, when abundance of the compound is too low, a sub-panel that is zoomed in is shown. We state this in the **Materials and Methods/LC-MS analysis** section in Supplementary information.

• *In Fig S25, panel A - it looks like one of your labels shown be “harvested 4 weeks after...”. You repeat “harvested 2 days after...”*

Response: We thank the reviewer for pointing this out. We have revised this typo (now in Extended Data Fig. 6 in our revised manuscript).

• *In Figure S29, it is difficult to tell the difference between blue and teal.*

Response: We have changed the teal to green (now in Extended Data Fig. 10 in our revised manuscript).

• *In Figure S38, could you label the peak as “9” - I think that would be helpful for readers.*

Response: We have labeled the peak as “prestrychnine 9” (now in the Supplementary Fig. 28 in the revised manuscript).

Editor Comments:

STATISTICS: Statistical analyses conform with Nature's guidelines.

REPRODUCIBILITY: Revised Reporting Summary and Editorial Policy Checklist is included.

LENGTH: The length is ca. 3000 words. There are 3 display items, with one of them taking up about half a page.

TITLE: The title is 26 characters.

SUMMARY PARAGRAPH: The summary paragraph is now labeled.

MAIN TEXT: A very brief introductory paragraph is included after the summary paragraph. We have no additional headings, though we can add them if the editor believes that they would improve the readability of the manuscript.

REFERENCES: We have 34 references.

FIGURE LEGENDS: Figure legends are now included at the end of the manuscript. Legends are less than 300 words. Statistical parameters are defined and the number of experiments is defined in Figure 3.

METHODS:

Since specialized synthetic procedures requiring chemical structures are part of the methods, the entire methods section is included as part of the Supplementary information, as specified in the instructions.

MAIN TEXT STATEMENTS:

All of the required statements have been included.

DATA AND CODE AVAILABILITY STATEMENTS:

The statements have been included. Please note that we are still waiting for a deposition code for the raw transcriptomic files. This will only take a few days.

DISPLAY ITEMS:

We have revised 3 figures accordingly. They are provide separately in doc format.

EXTENDED DATA:

We have now included 10 Extended Data figures that, in our judgement, are crucial for the main story of the manuscript. We have included these as 10 separate files. The legends are not included in the files, but are listed with the main text figure legends.

According to the Nature's Guideline, we put supplementary figures in the initial submission to Extended Data. Extended Data Fig. 1 in the revised manuscript is Supplementary Fig. 8 and 9 in the initial submission. Extended Data Fig. 2 is Supplementary Fig. 10. Extended Data Fig. 3 is Supplementary

Fig. 11 and 12. Extended Data Fig. 4 is Supplementary Fig. 15 and 16. Extended Data Fig. 5 is Supplementary Fig. 17. Extended Data Fig. 6 is Supplementary Fig. 25. Extended Data Fig. 7 is Supplementary Fig. 27. Extended Data Fig. 8 is Supplementary Fig. 28, 29, and 30. Extended Data Fig. 9 is Supplementary Fig. 32, 33, 34, and 35. Extended Data Fig. 10 is Supplementary Fig. 39.

SUPPLEMENTARY INFORMATION:

We have included Supplementary information (SI) with this manuscript.

Since the methods include synthetic procedures that include NMR data and chemical structures, we have included the methods as part of the SI, as stated in the formatting guidelines.

This manuscript has a large number of data files, in the format of LC chromatograms, phylogenetic trees, protein models, microscopy data, steady state kinetic data and mechanistic proposals that are not absolutely essential for following the main text, but validate some of the conclusions that we have drawn in the manuscript, with some of these being requested by the reviewers. These data realistically have to be presented in the form of figures. This seems to be consistent with the format that other Nature manuscripts in this general area have used. We are open to suggestions from the editor if there is a better way to include these data.

SOURCE DATA (GRAPHS):

We have included source data for relevant figures in the form of excel files. No more than one file per figure is provided.

Reviewer Reports on the First Revision:

Referees' comments:

Referee #1 (Remarks to the Author):

The authors have thoroughly addressed my comments. I have no additional comments.

Referee #2 (Remarks to the Author):

I appreciate the authors revisions to the manuscript. They have addressed all my comments, and I have no further feedback.

Author Rebuttals to First Revision:

25 May 2022

To the Editor,

We thank both of the reviewers for thoughtful comments throughout this process. A point-by-point response is below.

Referee #1:

The authors have thoroughly addressed my comments. I have no additional comments.

Response: Thank you for fair comments of the work during the process. We are happy that we were able to address your concerns.

Referee #2:

I appreciate the authors revisions to the manuscript. They have addressed all my comments, and I have no further feedback.

Response: Thank you for your thoughtful feedback throughout this process. We are glad to be able to answer your questions.